# CoPS: Empowering LLM Agents with Provable Cross-Task Experience Sharing

## Abstract

Sequential reasoning in agent systems has been significantly advanced by large language models (LLMs), yet existing approaches face limitations. Reflection-driven reasoning relies solely on knowledge in pretrained models, limiting performance in novel scenarios, while experience-assisted reasoning often depends on external experiences and lacks clear principles for selecting representative experiences. We address these limitations by proposing CoPS (**Cro**ss-Task Ex**p**erience **S**haring), a generalizable algorithm that enhances sequential reasoning by cross-task experience sharing and selection. In detail, CoPS leverages agents' experiences on previous tasks, selecting distribution-matched experiences via a provable pessimism-based strategy to maximize utility while minimizing risks from distribution shifts. Extensive experimental results on benchmarks like Alfworld, Webshop, and HotPotQA demonstrate that CoPS consistently outperforms state-of-the-art baselines, with superior sample efficiency suitable for resource-constrained scenarios. Theoretically, we show that the performance of our algorithm depends on both the quality of the pretrained LLM and the matching between the agent's task-dependent trial distribution and that generated by the LLM. Our work bridges the gap between existing sequential reasoning paradigms and validates the effectiveness of leveraging cross-task experiences, shedding light on the potential to improve agents' generalization and adaptability across diverse tasks. Our codes are released at this link.

## 1 Introduction

Burgeoning agent systems driven by advanced large language models (LLMs, (Devlin et al., 2019; Brown et al., 2020; OpenAI, 2023; Hu et al., 2024a)) have demonstrated remarkable capabilities in solving complex tasks through sequential reasoning (Qin et al., 2024; Hao et al., 2023; Huang et al., 2024; Chen et al., 2024b;a; Li et al., 2023a). These agent systems employ two typical sequential reasoning paradigms: reflection-driven reasoning and experience-assisted reasoning. Reflection-driven reasoning leverages a model's internal capabilities through methods such as reflection (Shinn et al., 2024), long-term rollouts (Zhou et al., 2023), or chain-of-thought (CoT) reasoning (Wei et al., 2022). While this approach capitalizes on the knowledge within the pre-trained model, it faces notable limitations. Specifically, relying solely on existing knowledge in the pre-trained model to generate rationales restricts the model's performance when encountering novel scenarios. Moreover, there is an increased risk of hallucinations, where internal reasoning may lead to plausible but incorrect responses (Huang et al., 2023). These challenges highlight the need for integrating external experiences to enhance the agent's sequential reasoning capabilities.

In contrast, experience-assisted sequential reasoning utilizes retrieval-based methods that enable the agent to interact with a memory bank of experiences, allowing the model to overcome knowledge cutoffs, personalize responses, and reduce hallucinations. However, these experiences are often manually curated or sourced from expert models (Raparthy et al., 2023), which is resource-intensive and poses scalability issues. Additionally, experience-assisted reasoning often lacks clear principles for selecting representative examples (Kagaya et al., 2024), potentially underutilizing the value of past experiences. These limitations bring us to a critical research question:

*Can agent systems enhance sequential reasoning by sharing and selecting cross-task experiences?*

> You are in the middle of a room. Looking quickly around you, you see a drawer 2, a shelf 5, a drawer 1, a shelf 4, a sidetable 1, a drawer 5, a shelf 6, a shelf 1, a shelf 9, a cabinet 2, a sofa 1, a cabinet 1, a shelf 3, a cabinet 3, a drawer 3, a shelf 11, a shelf 2, a shelf 10, a dresser 1, a shelf 12, a garbagecan 1, a armchair 1, a cabinet 4, a shelf 7, a shelf 8, a safe 1, and a drawer 4. Your task is to: put some vase in safe.

Figure 1: A brief illustration of CoPS, which fully leverages agents' cross-task experiences to enhance sequential reasoning by sharing and selecting distribution-matched experiences from previous task trajectories.

To address this question, we propose CoPS (**Cro**ss-Task Ex**p**erience **S**haring), a theoretically grounded algorithm that empowers agent systems through cross-task experience sharing and selection. CoPS demonstrates its generalizability by working effectively in both settings: utilizing fully external experiences in the *offline* setting and leveraging completely self-derived experiences in the *online* setting. By utilizing representative cross-task experiences, CoPS enables agents to improve performance on new, complex sequential reasoning tasks. Our key contributions are summarized as follows:

- We introduce CoPS, a method that fully leverages agents' cross-task experiences to enhance sequential reasoning by selecting distribution-matched experiences from previous task trajectories. Central to our approach is a theoretically grounded experience selection strategy based on the pessimism principle, which aims to maximize the utility of successful, representative experiences while minimizing risks associated with distribution shifts from out-of-distribution samples. Notably, CoPS is agnostic to the agent's base model, task type, experience sources, and implementation framework, making it easy-to-use and generalizable across various settings.

- Experimentally, we validate CoPS on key benchmarks such as Alfworld (Shridhar et al., 2020), Webshop (Yao et al., 2022a), and HotPotQA (Yang et al., 2018). CoPS consistently outperforms state-of-the-art experience-assisted reasoning approaches like RAP (Kagaya et al., 2024) and reflection-driven reasoning methods, like Reflexion (Shinn et al., 2024) and LATS (Zhou et al., 2023). Moreover, CoPS demonstrates superior sample efficiency compared to resource-intensive methods like LATS, making it highly suitable for resource-constrained scenarios. These results showcase CoPS's effectiveness in practical applications.

- Theoretically, we show that in both offline and online settings, the performance of our pessimism-based algorithm depends on both the quality of the pre-trained LLM and the matching between the cross-task experience distribution decided by the trials selected by the agent, and a task-dependent experience distribution denoted by the LLM. Our findings shed light on general strategies for designing efficient experience sharing and selction algorithms and offer a comprehensive understanding of CoPS's effectiveness across different scenarios.

**Notations** We denote by $[n]$ the set $\{1, \ldots, n\}$. For two positive sequences $\{a_n\}$ and $\{b_n\}$ with $n = 1, 2, \ldots$, we write $a_n = O(b_n)$ if there exists an absolute constant $C > 0$ such that $a_n \le Cb_n$ holds for all $n \ge 1$ and write $a_n = \Omega(b_n)$ if there exists an absolute constant $C > 0$ such that $a_n \ge Cb_n$ holds for all $n \ge 1$. We use $\widetilde{O}(\cdot)$ to further hide the polylogarithmic factors. We use $(x_i)_{i=1}^n$ to denote sequence $(x_1, ..., x_n)$, and we use $\{x_i\}_{i=1}^n$ to denote the set $\{x_1, ..., x_n\}$. We use $D_H(p, q) = \sqrt{1/2 \cdot \int (\sqrt{p} - \sqrt{q})^2}$ to denote the Hellinger distance. We use $D_{TV}(p, q) = 1/2 \cdot \int |p - q|$ to denote the Total variation distance. We use $\chi^2(p, q) = \int p^2/q - 1$ to denote

---

**Algorithm 1** COPS: Cross-Task Experience Sharing

---

**Require:** Language model $\text{LLM}(\cdot|\cdot)$, memory bank $\mathcal{D} = \{\tau_1, \ldots, \tau_n\}$, decoder $\text{Dec}$, distance metric $d$, memory size $k$, maximum sequence length $H$.

1: Receive initial state $s_1$, receive state-sampled experience $\tau^{s_1}$ through decoder $\tau^{s_1} \sim \text{Dec}(\cdot|s_1)$.

2: Set the probability $\widehat{p} \in \Delta(\mathcal{D})$ as in (2.3), which approximately maximizes the following:

$$\widehat{p} = \underset{p \in \Delta(\mathcal{D})}{\text{argmax}} \, \mathbb{E}_{\tau \sim p}[r(\tau) - d(\tau, \tau^{s_1})]. \tag{2.1}$$

3: Repeatedly retrieve trials $\tau^1, \ldots, \tau^k \sim \widehat{p}$.
4: Concate $\tau^1, \ldots, \tau^k$ into one trajectory $\mathcal{T} = \tau^1|\ldots|\tau^k$, set $h \leftarrow 1$.
5: **while** NOT SUCCESS and $h < H$ **do**
6:    Obtain action $a_h \sim \text{LLM}(\cdot|\mathcal{T}, s_h)$, set $s_{h+1} \leftarrow s_h|a_h$, $h \leftarrow h + 1$.
7: **end while**

---

the chi-square distance. For two sentences $a$ and $b$, we use $a|b$ to denote the sentence formed by concatenating $a$ and $b$.

## 2 METHODOLOGY

### 2.1 PRELIMINARY

We consider a sequential decision-making scenario, consisting of a task space $\texttt{M}$, a state space $\texttt{S}$, and an action space $\texttt{A}$. The state $s \in \texttt{S}$ is defined as a descriptive sentence representing the history of the current task. For example: "You are in the middle of a room. Please find a path to reach the apple." The action $a \in \texttt{A}$ is a solution to the task, such as: "Move right. The apple is on the table." The agent interacts with the environment through trials. At the beginning of each trial, a task $\mathbf{M}$ is randomly drawn from the task space, $\mathbf{M} \sim \mathbb{P}^{\texttt{M}}$. The agent then observes an initial state $s_1$, sampled from the initial state distribution, $s_1 \sim \mathbb{P}_0^{\mathbf{M}}$. At each step $h$, the agent makes a decision $a_h$ based on the current state $s_h$, and the next state is updated as $s_{h+1} = s_h|a_h$. The agent either successfully completes the task or continues generating actions until reaching the maximum number of interactions $H$ between the agent and the environment. We define an *experience* $\tau$ as a complete trial, i.e., $\tau = s_h$, where $h \leq H$ is the final step of the current trial. The reward $r(s_h)$ denotes how effectively the experience solves the task, with $0 \leq r(s_h) \leq 1$.

In this work, we assume access to a large language model (LLM) to assist in decision-making. We represent the LLM as $\text{LLM}(a|\cdot)$, a conditional distribution of actions given the input sequence.

### 2.2 PROPOSED METHOD

We introduce our proposed method, COPS, based on distribution matching. COPS operates on a trial-wise basis, making it suitable for both the *offline setting*, where the agent has access to an external static dataset containing experiences, and the *online setting*, where the agent gathers experiences through interactions with the environment. Suppose our agent is at the start of a trial with an initial state $s_1 \sim \mathbb{P}_0^{\mathbf{M}}$. We introduce the key components of COPS as follows.

**Memory Bank** The agent has access to a memory bank $\mathcal{D}$ containing experiences, either from a pre-collected dataset (offline) or from previous experiences (online). We do not impose restrictions on $\mathcal{D}$, meaning that experiences in $\mathcal{D}$ exhibit great diversity. Specifically, an experience $\tau \in \mathcal{D}$ may correspond to different tasks $\mathbf{M}$ or to varying solution strategies for the same task. Our goal is to develop a strategy for retrieving experiences from $\mathcal{D}$ that assist in decision-making for the current task.

**Cross-Task Experience Sharing** COPS utilizes an external module called the *decoder*, denoted as $\text{Dec}$ in Line 1. In general, the decoder outputs a task-dependent distribution of experiences conditioned on the initial state $s_1$, reflecting how the LLM would solve the task $\mathbf{M}$ associated with $s_1$ without explicit instructions. With the decoder's help, the agent's goal is to find a probability distribution $\widehat{p}$ over all experiences in $\mathcal{D}$ that satisfies:

$$\widehat{p} = \underset{p \in \Delta(\mathcal{D})}{\operatorname{argmax}} \, \mathbb{E}_{\tau \sim p}[r(\tau)] - d(p, \text{Dec}(\cdot|s_1)), \tag{2.2}$$

where $d$ is a metric over distributions. Intuitively, (2.2) is similar to the *pessimism principle*, commonly used in offline RL literature (Jin et al., 2021). The goal of $\widehat{p}$ is to maximize the expected reward while keeping the distribution close to the one decoded by Dec. Importantly, $\widehat{p}$ supports the cross-task setting, as it does not restrict its support to experiences from the same task as $s_1$. For a given in-context memory size $k$, CoPS repeatedly samples experiences $\tau^1, \ldots, \tau^k$ from $\widehat{p}$, as shown in Line 3.

**Execution Planning** Let $\mathcal{T} = \tau^1 | \ldots | \tau^k$ represent the *experience collection* containing $\tau^1, \ldots, \tau^k$. Starting from the initial state $s_1$, the agent executes actions step-by-step, where each action $a_h$ is drawn from the LLM's distribution, conditioned on both the experience collection and the current state:

$$a_h \sim \text{LLM}(\cdot | \mathcal{T}, s_h).$$

In the online setting, after completing a trial, the agent updates the memory bank $\mathcal{D}$ by adding the new experience for future use.

**Implementation Details** Here we discuss several implementation details for CoPS. *First*, in practice, directly computing the distance $d(p, \text{Dec}(\cdot|s_1))$ between distributions in (2.2) is computationally intractable. Therefore, we use an empirical approximation to translate the distance between distributions into the distance between experiences drawn from those distributions, as shown in (2.1). *Second*, we specify the choice of Dec. The decoder outputs an experience $\tau^{s_1}$ from $\mathcal{D}$ that starts with the same initial state $s_1$. If multiple such experiences exist, we select the most recent one. This $\tau^{s_1}$ naturally reflects the behavior of the LLM for solving the task starting from $s_1$ without intervention. *Third*, we discuss how to approximately solve (2.1) since enumerating all possible distributions in $\Delta(\mathcal{D})$ is computationally inefficient. Specifically, we define the distance function $d$ and approximately solve $\widehat{p}$ as follows:

$$d(\tau, \tau') := c \cdot \cos(e(\tau), e(\tau')), \quad \widehat{p}(\tau) \propto r(\tau) \cdot \exp(-d(\tau, \tau^{s_1})), \tag{2.3}$$

where $c \geq 0$ is a hyperparameter, "cos" denotes the cosine function, and $e$ is an embedding function that maps a language sentence to a high-dimensional Euclidean space. In practice, we use $e$ as a language embedding model (e.g., gte-Qwen2 7b (Li et al., 2023b)). This approach favors selecting successful experiences from $\mathcal{D}$ with probabilities proportional to the inverse distance from the current initial state $s_1$. The hyperparameter $c$ in (2.3) controls the influence of relative distances: when $c = 0$, the method uniformly samples successful experiences from $\mathcal{D}$, and as $c \to \infty$, it deterministically selects the experience closest to $\tau^{s_1}$.

## 3 EXPERIMENTS SETUP

In this section, we present our experimental study evaluating the practical performance of CoPS on real-world LLMs, specifically the Llama 3.1 models (Dubey et al., 2024). Our results show that CoPS achieves state-of-the-art (SOTA) performance in both task success rate and sample efficiency, surpassing existing baselines to the best of our knowledge. A detailed description of our prompt formulation is provided in Appendix L. Notably, CoPS is both simple to implement and generalizable across different environments: for each trial, the selected experiences are straightforwardly added to the prompts, requiring no manual modifications.

This prompting strategy offers two distinct advantages: first, it significantly boosts sequential reasoning performance by incorporating cross-task experiences, outperforming reflection-driven methods like Reflexion. Second, the prompts across trials share a substantial prefix, which maximizes the effectiveness of prefix-caching mechanisms in modern LLM-serving systems (Zheng et al., 2023), leading to significant efficiency improvements over RAP (Kagaya et al., 2024).

**Benchmarks** We evaluate our algorithms on three representative benchmarks: **Alfworld** (Shridhar et al., 2020), **Webshop** (Yao et al., 2022a), and **HotPotQA** (Yang et al., 2018). In these benchmarks, agents strive to solve problems in limited number of trials, enabling cross-trial and cross-task experience sharing. In Alfworld, agents are provided with a specific task description within a simulated household environment, interacting through predefined actions and receiving feedback in the form

of textual descriptions. In Webshop, the agent must locate a product that matches user specifications from a catalog of over one million items, interacting with the HTML page and search engine while receiving limited product information per trial. In HotPotQA, the agent answers complex questions requiring specific knowledge, using Wikipedia to retrieve relevant articles. In all benchmarks, the reward function $r(\tau)$ is defined as 1 if the agent successfully completes the task and 0 otherwise.

**LLM Selection** We conduct our entire experiment with the widely-used Llama 3.1 series of models (Dubey et al., 2024), in consideration of their superior benchmark performance and the sustainability of open-weight LLM ecosystems. Specifically, our experiments are conducted with Llama 3.1 8b Instruct and Llama 3.1 70b Instruct on NVIDIA A6000 and A100 GPUs. We use gte-Qwen2 7b Instruct (Li et al., 2023b) as our embedding model. We use SGLang (Zheng et al., 2023) as our LLM-serving engine for its SOTA serving performance and prefix-caching mechanism.

**Baselines** We compare CoPS with three representative baselines: **Reflexion** (Shinn et al., 2024), **RAP** (Kagaya et al., 2024), and **LATS** (Zhou et al., 2023). In Reflexion, the agent try to solve the task in each environment over multiple trials until it succeeds. After each failed attempt, the LLM agent reflects on its unsuccessful trajectory and saves this reflection in its memory. For each subsequent trial, the agent is provided with up to three recent reflections from the same task. In RAP, at each stage within a trial, the agent is presented with the top-$k$ search results of trajectory fragments as in-context demonstrations. In LATS, the agent utilizes a tree-structured search to explore multiple reasoning and action rationales at each trial. When it encounters failed rationales, the agent generates reflections on its mistakes and integrates these insights into its decision-making process for future trials.

## 4    RESULTS AND ANALYSIS

In this section, we demonstrate that CoPS outperforms all baselines across all benchmarks and model sizes, considering both sample efficiency and task success rate. Detailed performance illustrations over multiple trials are presented in Figure 2.

**Alfworld Benchmark** Table 2 and Figures 2(a), 2(d) illustrate the comparison between CoPS, Reflexion, and RAP on the Alfworld benchmark. The values represent the success rate after 10 trials across 134 tasks. When using the smaller Llama 3.1 8b model, CoPS reaches a success rate of 94%, significantly surpassing both Reflexion (86%) and RAP (70%). This result is particularly noteworthy because Reflexion requires the much larger Llama 3.1 70b model to achieve similar performance, highlighting superior effectiveness of CoPS. This demonstrates CoPS's ability

Table 1: Performance comparison of Reflexion, RAP, and CoPS on Alfworld benchmark using Llama3.1 8b and 70b models.

| Algorithm | Performance | |
|---|---|---|
| | **Llama3.1 8b** | **Llama3.1 70b** |
| Reflexion[1] | 86 | 94 |
| RAP | 70 | 93 |
| **CoPS** | **94** | **100** |

to achieve state-of-the-art performance even with limited computational resources and less capable models, offering a clear advantage over other algorithms. Furthermore, when scaling to the larger Llama 3.1 70b model, CoPS achieves a perfect success rate of 100%. These results emphasize that CoPS scales effectively, consistently outperforming the baselines across model sizes. Although RAP also leverages an in-context demonstrations retrieval mechanism, it lacks an effective experiences selection algorithm, thus noticeably underperforms CoPS. Additionally, it is important to note that RAP manually splits the agent's planning trajectory into multiple stages for each trial, and these split methods are specific to each benchmark and must be manually tailored. This significantly increases implementation complexity and introduces scalability issues. In contrast, CoPS efficiently reuses successful experiences by directly placing them in the prompts, without requiring benchmark-specific modifications, making it a more practical and flexible solution. As a result, CoPS not only surpasses the baselines in performance but also offers out-of-the-box usability by eliminating the need for manual intervention.

---

[1]The original codebase of Reflexion struggles to perform on most tasks with the smaller Llama3.1 8b model. This is primarily because the model tends to repeat the same action, leading to task failure. To mitigate this, we introduced a resampling mechanism to enhance Reflexion performance, which activates when the model begins to repeat actions. This modification significantly improved Reflexion's performance.

**Webshop Benchmark**[2] Table 2 and Figures 2(b), 2(e) compare the performance of CoPS with all baseline algorithms on the Webshop benchmark, measured in terms of success rate. The values indicate the success rate over 50 products, with each algorithm evaluated through 10 trials per product. For the smaller Llama 3.1 8b model, CoPS achieves a success rate of 50%, outperforming the next best competitor, RAP, by a substantial absolute improvement of 8%. When scaling to the larger Llama 3.1 70b model, the performance gain of CoPS becomes even more pronounced, with a success rate of 56%. This marks a 14% absolute improvement over RAP.

To ensure a fair comparison across the baselines, we modified the LATS baseline by reducing the width of the search tree and limiting the number of trajectory iterations. This adjustment ensures that the running time spent on each baseline is approximately equal. Even with these changes, LATS still exhibits significantly lower sample efficiency. Specifically, the total number of tokens generated by Llama 3.1 8b in LATS (1,555,365 tokens) is nearly five times greater than that in CoPS (314,336 tokens). Further details can be found in Table 4 in Appendix C. This discrepancy in token usage highlights the inefficiency of current search-tree-based algorithms. In contrast, CoPS demonstrates much better efficiency and performance under the same inference constraints.

Table 2: Performance comparison of Reflexion, RAP, LATS, and CoPS on Webshop benchmark using Llama3.1 8b and 70b models.

| Algorithm | Performance | |
|---|---|---|
| | **Llama3.1 8b** | **Llama3.1 70b** |
| Reflexion | 30 | 30 |
| RAP | 42 | 42 |
| LATS | 24 | 32 |
| **CoPS** | **50** | **56** |

**HotPotQA Benchmark** Table 3 and Figures 2(c), 2(f) illustrate the comparison between CoPS, Reflexion, and LATS on the HotPotQA benchmark, conducted on 100 question-answering (QA) tasks. The values in the table represent the success rates, with each algorithm being tested over 10 trials. As evidenced by the results, CoPS consistently achieves superior performance relative to both Reflexion and LATS across all model sizes. The advantage of CoPS is particularly evident when using the smaller Llama 3.1 8b model, where CoPS achieves a success rate of 63%, outperforming Reflexion and LATS by substantial absolute improvements of 7% and 8%, respectively. Moreover, even when scaled up to the larger Llama 3.1 70b model, CoPS continues to gain stronger performance. In this setting, CoPS reaches a success rate of 65%, surpassing Reflexion by 4% and LATS by 1%. Note that both Reflexion and LATS baselines demonstrate a significant performance gap when shifting from smaller to larger model, while the results for CoPS is relatively consistent and maintains the performance edge throughout different sizes of models. This demonstrates that CoPS's principled cross-task experience sharing mechanism also excels in tasks requiring complex reasoning.

Table 3: Performance comparison of Reflexion, LATS, and CoPS on HotPotQA benchmark using Llama3.1 8b and 70b models.

| Algorithm | Performance | |
|---|---|---|
| | **Llama3.1 8b** | **Llama3.1 70b** |
| Reflexion | 56 | 61 |
| LATS | 55 | 64 |
| **CoPS** | **63** | **65** |

**Conclusion**[3] Our experiments across Alfworld, Webshop, and HotPotQA demonstrate that CoPS consistently outperforms state-of-the-art baselines in both task success rate and sample efficiency. Notably, CoPS achieves superior performance even with smaller models like Llama 3.1 8b, highlighting its efficiency and practicality for resource-constrained scenarios. These results validate the effectiveness of leveraging principled cross-task experiences sharing through our theoretically grounded selection strategy, confirming that CoPS enhances sequential reasoning capabilities across diverse tasks and model sizes.

---

[2]We observed that scaling up the model sizes for Reflexion and RAP on the Webshop benchmark did not result in significant improvements. This observation aligns with the original findings of Reflexion (Shinn et al., 2024, Appendix B.1) and RAP (Kagaya et al., 2024, Table 2), which suggest that these models tend to converge on local minima that require highly creative strategies to overcome.

[3]We also conduct ablation studies on tuning key hyperparameters of CoPS in Appendix B, providing practical guidance for hyperparameter selection for optimal performance of CoPS.

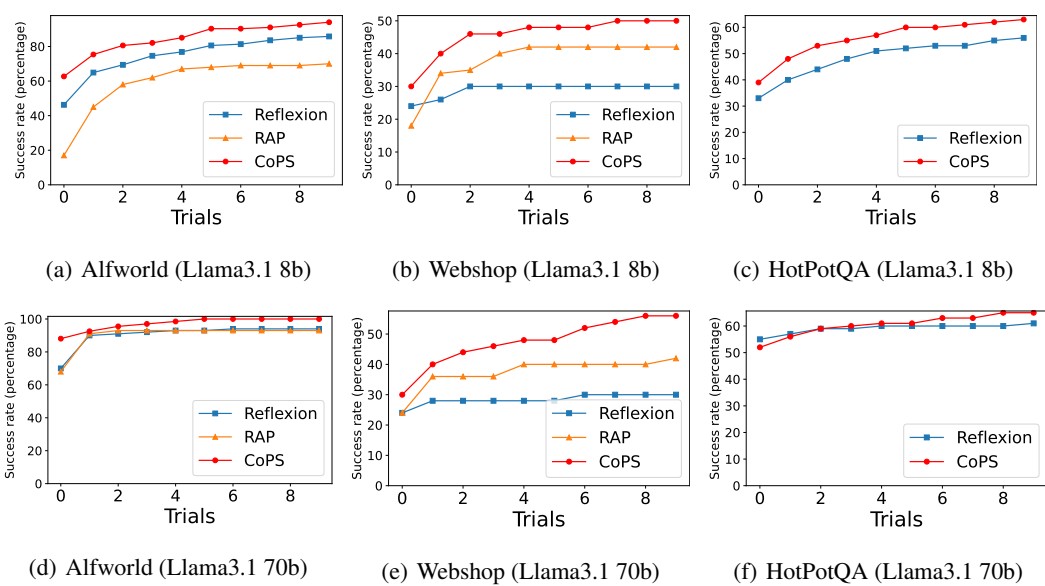

Figure 2: Comparative evaluation of CoPS, Reflexion, RAP, and LATS across three benchmarks: Alfworld, Webshop, and HotPotQA. The figures illustrate the success rates for both the smaller Llama 3.1 8b and larger Llama 3.1 70b models, averaged over 10 trials.

## 5 THEORETICAL FRAMEWORK OF EXPERIENCE-ASSISTED AGENTS

In this section, we develop the theoretical framework to demonstrate the effectiveness of CoPS. For simplicity, we analyze our algorithm in a bandit setting, where the maximum number of steps for each experience is $H = 1$. Slightly different from the formulation in Section 2, we define an experience as $\tau = s|a|r$, consisting of an initial state $s$, an action $a$, and its reward $r = r(s, a)$.

We introduce additional notations for clarity in our analysis. Let $\mathcal{T} = \tau_1|\tau_2|\ldots$ denote the experience collection. The length of $\mathcal{T}$ is denoted by $|\mathcal{T}|$, i.e., $\mathcal{T} = (\tau_1, ..., \tau_{|\mathcal{T}|})$. We use $\mathcal{T}_t$ to represent the first $t$ steps of the experience collection, i.e., $\mathcal{T}_t = \tau_1|\ldots|\tau_t$. For any experience collection $\mathcal{T}$, we assume $|\mathcal{T}| \leq T$. We define T as the space of all trajectories, and $T_t$ as the space of trajectories of length $t$. We denote a general algorithm as $\mathrm{Alg}(\cdot|\cdot, \cdot, \cdot) : \mathtt{M} \times \mathtt{T} \times \mathtt{S} \to \Delta(\mathtt{A})$, which takes as input a task $\mathbf{M} \in \mathtt{M}$, an experience collection $\mathcal{T} \in \mathtt{T}$, and a state $s \in \mathtt{S}$, and outputs a distribution over actions $a \in \mathtt{A}$. Note that some algorithms may not use the task $\mathbf{M}$ as input, in which case we write $\mathrm{Alg}(\cdot|\cdot, \cdot)$. We denote $\mathbb{P}_t^{\mathbf{M},\mathrm{Alg}}$ as the distribution over the first $t$ steps of an experience collection under task $\mathbf{M}$ and algorithm Alg. For an algorithm Alg that takes $\mathbf{M}, \mathcal{T}, s$ as input, we define its *posterior average* as $\overline{\mathrm{Alg}}(\cdot|\mathcal{T}, s) = \mathbb{E}_{\mathbf{M} \sim \mathbb{P}^{\mathbb{M}}(\cdot|\mathcal{T}'=\mathcal{T}, s'=s)}[\mathrm{Alg}(\cdot|\mathbf{M}, \mathcal{T}', s')]$, which is the best Bayesian approximation of Alg given the experience collection $\mathcal{T}$ and current state $s$.

### 5.1 LLM PRETRAINING

We begin by describing the pretraining process for the LLM. Let $\mathrm{Alg}_{\widehat{\boldsymbol{\theta}}}(\cdot|\mathcal{T}, s) : \mathtt{T} \times \mathtt{S} \to \Delta(\mathtt{A})$ represent an LLM agent that outputs a distribution over A, where $\widehat{\boldsymbol{\theta}} \in \boldsymbol{\Theta}$ is the parameter of the LLM, $\boldsymbol{\Theta}$ denotes the whole parameter space. We assume that there exists a pretraining dataset $\mathcal{D}_{\mathrm{pre}} = \{\mathcal{T}^1, \ldots, \mathcal{T}^{n_{\mathrm{pre}}}\}$, with $|\mathcal{T}^i| = T - 1$. Following the pretraining setup in Lin et al. (2023), we assume two algorithms: a *context algorithm*, $\mathrm{Alg}^C(\cdot|\cdot, \cdot) : \mathtt{T} \times \mathtt{S} \to \Delta(\mathtt{A})$, and an *expert algorithm*, $\mathrm{Alg}^E(\cdot|\cdot, \cdot, \cdot) : \mathtt{M} \times \mathtt{T} \times \mathtt{S} \to \Delta(\mathtt{A})$. In general, the context algorithm provides a "natural" action based on the experience collection and current state, while the expert algorithm provides a more informed action, given the task information, experience collection, and current state. Since the expert algorithm has access to task information $\mathbf{M}$, it typically produces better actions than the context algorithm.

We now describe the pretraining process. To generate an experience collection $\mathcal{T} = \tau_1|\ldots|\tau_{T-1} \in \mathcal{D}_{\mathrm{pre}}$, we first sample a task $\mathbf{M} \sim \mathbb{P}^{\mathbb{M}}$. For each experience $\tau_i$, the state is sampled from the initial

state distribution $s_i \sim \mathbb{P}_0^{\mathbf{M}}$, the action is sampled using the context algorithm $a_i \sim \mathrm{Alg}^C(\cdot|\mathcal{T}_{i-1}, s_i)$, and the reward is given by $r_i = r(s_i, a_i)$. After generating the experience collection, we collect expert feedback $\bar{a}_1, \ldots, \bar{a}_{T-1}$ for each step of $\mathcal{T}$, using the expert algorithm, where $\bar{a}_i \sim \mathrm{Alg}^E(\cdot|\mathbf{M}, \mathcal{T}_{i-1}, s_i)$. Repeating this process $n_{\mathrm{pre}}$ times produces the trajectories $\mathcal{T}^i$ and expert actions $\bar{a}_1^i, \ldots, \bar{a}_{T-1}^i$ for $i \in [n_{\mathrm{pre}}]$. Finally, we pretrain the LLM $\mathrm{Alg}_{\widehat{\boldsymbol{\theta}}}$ by solving the following maximum likelihood estimation problem:

$$\widehat{\boldsymbol{\theta}} \leftarrow \underset{\boldsymbol{\theta} \in \boldsymbol{\Theta}}{\arg\max} \sum_{i=1}^{n_{\mathrm{pre}}} \sum_{t=1}^{T} \log \mathrm{Alg}_{\boldsymbol{\theta}}(\bar{a}_t^i | \mathcal{T}_{t-1}^i, s_t^i).$$

For the remainder of this paper, we use $\mathrm{Alg}_{\widehat{\boldsymbol{\theta}}}$ to represent our LLM. Below, we present several standard assumptions for analyzing $\mathrm{Alg}_{\widehat{\boldsymbol{\theta}}}$.

**Definition 5.1** (Lin et al. 2023)**.** Let $\boldsymbol{\Theta}$ be the set of parameters of the LLM, $\mathrm{Alg}_{\boldsymbol{\theta}}$. We call $\boldsymbol{\Theta}_0 \subseteq \boldsymbol{\Theta}$ a $\rho$-cover of $\boldsymbol{\Theta}$ with respect to $\mathrm{Alg}_{\boldsymbol{\theta}}$ if, for any $\boldsymbol{\theta} \in \boldsymbol{\Theta}$, there exists $\boldsymbol{\theta}_0 \in \boldsymbol{\Theta}_0$ such that

$$\forall s \in \mathtt{S}, t \in [T], \mathcal{T} \in \mathtt{T}_{t-1}, \| \log \mathrm{Alg}_{\boldsymbol{\theta}}(\cdot|\mathcal{T}, s) - \log \mathrm{Alg}_{\boldsymbol{\theta}_0}(\cdot|\mathcal{T}, s)\|_{\infty} \leq \rho.$$

We denote $\mathcal{N}(\rho) = |\boldsymbol{\Theta}_0|$ as the $\rho$-covering number of $\mathrm{Alg}_{\boldsymbol{\theta}}$.

Next assumption assumes that, the best approximation between the trained LLM and the posterior average of the expert algorithm, $\overline{\mathrm{Alg}^E}$, can be bounded by some constant.

**Assumption 5.2** (Lin et al. 2023)**.** There exists $\boldsymbol{\theta}^* \in \boldsymbol{\Theta}$ and a *model capacity error* $\epsilon_{\mathrm{real}} > 0$ such that

$$\forall t \in [T], \log \mathbb{E}_{\mathbf{M} \sim \mathbb{P}^{\mathbf{M}}, s \sim \mathbb{P}_0^{\mathbf{M}}, \mathcal{T} \sim \mathbb{P}_{t-1}^{\mathbf{M}, \mathrm{Alg}^C}, \bar{a} \sim \overline{\mathrm{Alg}^E}(\cdot|\mathcal{T}, s)} \left[ \frac{\overline{\mathrm{Alg}^E}(\bar{a}|\mathcal{T}, s)}{\mathrm{Alg}_{\boldsymbol{\theta}^*}(\bar{a}|\mathcal{T}, s)} \right] \leq \epsilon_{\mathrm{real}}.$$

Finally, we make assumptions for the decoder $\mathtt{Dec}$ introduced in Algorithm 1. We assume access to a class of decoders $\mathtt{Dec}_t : \mathtt{S} \to \Delta(\mathtt{T}_t)$ that maps the state $s$ to a distribution over the space of $t$ number of experiences, capable of estimating the distribution $\mathbb{P}_t^{\mathbf{M}, \mathrm{Alg}^C}(\mathcal{T})$, which represents the task-dependent experience distribution offered by LLM.

**Assumption 5.3.** For the decoder $\mathtt{Dec}_t : \mathtt{S} \to \Delta(\mathtt{T}_t)$, there exists a *decoder coefficient* $C_{\mathtt{Dec}} > 1$ such that for any $t \in [T], \mathcal{T} \in \mathtt{T}_{t-1}, \mathbf{M} \in \mathtt{M}$ and $s \sim \mathbb{P}_0^{\mathbf{M}}$, we have

$$\frac{1}{C_{\mathtt{Dec}}^2} \leq \frac{\mathtt{Dec}_{t-1}(\mathcal{T}|s)}{\mathbb{P}_{t-1}^{\mathbf{M}, \mathrm{Alg}^C}(\mathcal{T})} \leq C_{\mathtt{Dec}}^2.$$

## 5.2 Algorithm Analysis

We consider the same offline setting as in Section 2. Suppose we have an offline dataset $\mathcal{D}$, and the agent is given an initial state $s$. We formalize the experience selection problem as a distribution selection problem, where the agent has access to a candidate set of distributions, denoted by $\mathcal{P} = \{\mathbb{P}^1(\cdot|\cdot, \cdot), \ldots, \mathbb{P}^{|\mathcal{P}|}(\cdot|\cdot, \cdot)\} \subseteq 2^{\mathtt{T}_{T-1} \times \mathtt{S} \to \Delta(\mathtt{T}_{T-1})}$. Each element in this set represents a mapping from the dataset $\mathcal{D}$ and the current state $s$ to a distribution over trajectories $\mathcal{T}$ of length $T - 1$. In general, each $\mathbb{P}^i$ can be interpreted as the distribution over all possible combinations of $T - 1$ experiences from the dataset $\mathcal{D}$. The agent's goal is to select a distribution $\widehat{\mathbb{P}}^s$ from $\mathcal{P}$ that minimizes the suboptimality gap, which quantifies the performance difference between the best possible strategy and the strategy selected by the agent, as measured by the expert algorithm:

$$\mathrm{SubOpt}(\widehat{\mathbb{P}}^s) := \mathbb{E}_{\mathbf{M} \sim \mathbb{P}^{\mathbf{M}}, s \sim \mathbb{P}_0^{\mathbf{M}}} \left[ \max_{\widehat{\mathbb{P}} \in \mathcal{P}} \mathbb{E}_{\mathcal{T} \sim \widehat{\mathbb{P}}, a \sim \overline{\mathrm{Alg}^E}(\cdot|\mathcal{T}, s)} r(s, a) - \mathbb{E}_{\mathcal{T} \sim \widehat{\mathbb{P}}^s, a \sim \overline{\mathrm{Alg}^E}(\cdot|\mathcal{T}, s)} r(s, a) \right]. \tag{5.1}$$

We propose OFFLINECOPS in Algorithm 2, which is an experience collection-based version of COPS. The core idea of OFFLINECOPS mirrors that of COPS: the agent seeks to find experience collection that maximize the reward while minimizing the distributional shift from the experience collection of the current task, denoted by LLM. Given the test state $s$, OFFLINECOPS first runs the decoder to obtain a distribution $\mathtt{Dec}_{T-1}(\cdot|s)$, which approximates $\mathbb{P}_{t-1}^{\mathcal{M}, \mathrm{Alg}^C}$. Then, OFFLINECOPS

---

**Algorithm 2** OFFLINECOPS

---

**Require:** LLM $\text{Alg}_{\widehat{\boldsymbol{\theta}}}(\cdot|\cdot,\cdot)$, candidate experience collection distribution $\mathcal{P}$, pretraining error parameter $\epsilon_{\text{pretrain}}$, task decoder $\text{Dec}$, offline dataset $\mathcal{D}$.

1: Receive test state $s$, decode the distribution $\text{Dec}_{T-1}(\cdot|s)$.

2: Select $\widehat{\mathbb{P}}^s$ from $\mathcal{P}$ that maximizes the following:

$$\widehat{\mathbb{P}}^s = \underset{\widehat{\mathbb{P}}\in\mathcal{P}}{\text{argmax}}\,\mathbb{E}_{\substack{\mathcal{T}\sim\widehat{\mathbb{P}}(\cdot|\mathcal{D},s),\\a\sim\text{Alg}_{\widehat{\boldsymbol{\theta}}}(\cdot|\mathcal{T},s)}}\,r(s,a) - \epsilon_{\text{pretrain}}\sqrt{1+\chi^2(\widehat{\mathbb{P}}(\cdot|\mathcal{D},s),\text{Dec}_{T-1}(\cdot|s))}. \qquad (5.2)$$

3: Generate $\mathcal{T}^s \sim \widehat{\mathbb{P}}^s$ and obtain $a \sim \text{Alg}_{\widehat{\boldsymbol{\theta}}}(\cdot|\mathcal{T}^s,s)$.

---

applies the *pessimism principle*, as in (2.2). The selected distribution $\mathbb{P}^* \in \mathcal{P}$ aims to identify a distribution that produces an experience collection which maximizes the reward given the actions provided by the LLM, while staying close to the decoded distribution $\text{Dec}_{T-1}(\cdot|s)$. To measure the distributional distance, we employ the $\chi^2$-distance. Similar to the hyperparameter $c$ in COPS, OF-FLINECOPS introduces a hyperparameter $\epsilon_{\text{pre}}$ to balance the trade-off between maximizing reward and satisfying the regularity condition imposed by $\text{Dec}_{T-1}(\cdot|s)$.

We have the following theorem to characterize the performance of OFFLINECOPS.

**Theorem 5.4.** By setting

$$\epsilon_{\text{pretrain}} = C_{\text{Dec}}T \cdot \sqrt{5 \cdot T\log(\mathcal{N}(1/(n_{\text{pre}}T)^2)T) \cdot n_{\text{pre}}^{-1} + T\epsilon_{\text{real}}},$$

and denote $\mathbb{P}^{*,s} = \text{argmax}_{\widehat{\mathbb{P}}\in\mathcal{P}}\,\mathbb{E}_{\mathcal{T}\sim\widehat{\mathbb{P}}(\cdot|\mathcal{D},s),a\sim\overline{\text{Alg}^E}(\cdot|\mathcal{T},s)}\,r(s,a)$, we have the following bound with probability at least $1-2/T$:

$$\text{SubOpt}(\widehat{\mathbb{P}}^s) \leq 2C_{\text{Dec}}\epsilon_{\text{pretrain}}\mathbb{E}_{\mathbf{M}\sim\mathbb{P}^{\mathbf{M}},s\sim\mathbb{P}_0^{\mathbf{M}}}\sqrt{1+\chi^2(\mathbb{P}^{*,s}(\cdot|\mathcal{D},s),\mathbb{P}_{T-1}^{\mathbf{M},\text{Alg}^C}(\cdot))}.$$

*Proof.* See Appendix A.1. $\qquad\square$

Theorem 5.4 provides several insights into why COPS achieves superior performance and how experience selection should be tailored under different circumstances:

- The final suboptimality gap of the selected distribution $\mathbb{P}^{*,s}$ depends on the decoder coefficient $C_{\text{Dec}}$ and the pretraining error parameter $\epsilon_{\text{pre}}$. This implies that for a more powerful LLM, the selected experience distribution $\mathbb{P}^{*,s}$ will be closer to the optimal distribution. Meanwhile, the dependence of $\mathbb{P}_{T-1}^{\mathbf{M},\text{Alg}^C}$ suggests that the task-dependent experience collection distribution offered by LLM serves as a strong regularizer to select the optimal retrieval strategy.

- The optimal choice of the pretraining error parameter $\epsilon_{\text{pre}}$ is influenced by the decoder coefficient $C_{\text{Dec}}$, the number of pretraining trajectories in the pretraining set $n_{\text{pre}}$, and the model capacity error $\epsilon_{\text{real}}$. In general, for a more powerful LLM, where $n_{\text{pre}}$ is large and $\epsilon_{\text{real}}$ is small, our theorem suggests that the agent should focus more on aligning the selected experience collection distribution $\mathbb{P}^{*,s}$ with the decoder distribution $\text{Dec}$. This aligns with our observations in Section B, where smaller models, such as LLaMA 3.1 8b, are more sensitive to the choice of the hyperparameter $c$.

Due to the space limit, we leave the algorithm and analysis for the online setting to Appendix A.2.

# 6 RELATED WORK

## 6.1 LLM-POWERED AGENTS

In recent years, there has been a significant surge in research focused on LLM-powered agents (Chen et al., 2024b;a; Chan et al., 2023). React (Yao et al., 2022b) laid the foundation for much of the subsequent work on LLM agents, particularly those based on in-context learning (ICL). The most relevant studies to COPS include (Shinn et al., 2024; Kagaya et al., 2024; Zhou et al., 2023; Raparthy et al., 2023). In (Kagaya et al., 2024), a retrieval process for selecting in-context demonstrations was proposed. However, their approach depends on frequent embedding queries during

the planning stage, leading to inefficiency issues even in smaller LLM settings. Additionally, RAP manually splits the agent's planning trajectory into multiple stages for each trial, with benchmark-specific tailoring, significantly increases implementation complexity and raises scalability concerns. (Zhou et al., 2023) introduced a Tree-of-Thought (ToT) approach (Yao et al., 2024), incorporating backpropagation and a valuation process. However, their approach demonstrated poor sample efficiency, making it less suited for real-world agent settings where opportunities for trial and error are limited. Similarly, (Liu et al., 2023) integrated value-based search into a theoretical framework, but faced similar challenges with sample efficiency. (Feng et al., 2024) explored fine-tuning for specific LLM agent tasks, achieving good performance but with high computational costs. Lastly, (Raparthy et al., 2023) utilized high-quality experiences as ICL demonstrations for sequential reasoning. Although achieving remarkable performance, these experiences are introduced from external RL systems, which is resource-intensive and poses scalability issues. O3D (Xiao et al., 2024) is also highly related to CoPS, which introduces an offline learning framework that leverages skill discovery and knowledge distillation to enhance cross-task generalization without requiring fine-tuning, excelling in offline settings and diverse domains. In contrast, CoPS addresses cross-task experience selection using a pessimism-based strategy to mitigate distribution shifts, enabling dynamic adaptation and superior sample efficiency even in resource-constrained environments.

## 6.2 IN-CONTEXT DEMONSTRATIONS SELECTION

The selection of demonstrations for ICL has been widely studied. (Wang et al., 2024b) approached in-context demonstration selection from a Bayesian perspective, explicitly constructing a latent variable for the selection process. However, their analysis did not account for the pre-trained knowledge distribution, and their results were primarily empirical. (Yan et al., 2023) investigated the impact of repetition in in-context demonstrations, conducting controlled experiments to assess how repetitions in pre-trained knowledge influence results. (Scarlatos & Lan, 2023) developed a reinforcement learning framework to select in-context examples, while (Voronov et al., 2024) examined the impact of prompt formatting on in-context learning performance. Additionally, (Shum et al., 2023) introduced an automatic CoT augmentation and selection method for ICL example datasets. (Hu et al., 2024b) analyzed the scaling of in-context demonstrations from a theoretical standpoint, deriving general statistical bounds while accounting for pre-training errors. However, their focus was primarily on CoT in general ICL settings, not on the specific challenges faced by LLM agents interacting with environments and requiring feedback for optimization.

## 6.3 THEORY OF AGENTS

Several works have advanced the theoretical understanding of LLM agents. (He et al., 2024) explored the statistical theory of LLM agents through the lens of Bayesian aggregated imitation learning. (Lin et al., 2023) provided a theoretical analysis of transformers within the context of in-context reinforcement learning. (Wang et al., 2024a) examined the training and generalization of transformers for sequential reasoning, drawing parallels between transformer behavior and online learning algorithms. (Sumers et al., 2023) offered a cognitive perspective on LLM agents, while (Park et al., 2024) investigated the regret of LLM agents in sequential reasoning tasks, contributing both theoretical and empirical insights that inform CoPS's development.

## 7 CONCLUSION

In this paper, we introduced CoPS (**Cro**ss-Task Ex**p**erience **S**haring), a theoretically grounded algorithm that empowers agent systems with cross-task experiences sharing. Using a pessimism-based strategy to select relevant experiences, CoPS maximizes utility while minimizing the risks of distribution shifts. Our experiments on benchmarks like Alfworld, Webshop, and HotPotQA demonstrate that CoPS outperforms state-of-the-art methods in both success rates and sample efficiency. Theoretically, we show that our algorithm's performance depends on the LLM's pre-trained quality and the matching between the cross-task experience distribution decided by the trials selected by the agent, and a task-dependent experience distribution denoted by the LLM, providing insights for improving experience retrieval methods.

---

[4]We demonstrate the limitations of CoPS in Appendix J due the page constraints.

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

# A  ADDITIONAL DETAILS IN SECTION 5

## A.1  PROOF OF THEOREM 5.4

We prove Theorem 5.4 here. First, we need the following lemmas.

**Lemma A.1** (Lemma 20, Lin et al. 2023). With probability at least $1 - \delta$, we have

$$\mathbb{E}_{\mathbf{M} \sim \mathbb{P}^{\mathbf{M}}, s \sim \mathbb{P}_0^{\mathbf{M}}, \mathcal{T} \sim \mathbb{P}_{T-1}^{\mathbf{M}, \mathrm{Alg}^C}} \left[ \sum_{t=1}^{T} \mathrm{D_H}^2 (\overline{\mathrm{Alg}^E}(\cdot | \mathcal{T}_{t-1}, s), \mathrm{Alg}_{\widehat{\theta}}(\cdot | \mathcal{T}_{t-1}, s)) \right]$$

$$\leq 5 \cdot \frac{T \log(\mathcal{N}(1/(n_{\mathrm{pre}} T)^2) T / \delta)}{n_{\mathrm{pre}}} + T \epsilon_{\mathrm{real}},$$

where the covering number $\mathcal{N}$ is defined in Definition 5.1, $\epsilon_{\mathrm{real}}$ is defined in Assumption 5.2.

Next lemma is used to provide a per-state guarantee for the generalization error.

**Lemma A.2.** Let event $\mathcal{E}$ be defined as

$$\mathbb{E}_{\mathcal{T} \sim \mathbb{P}_{T-1}^{\mathbf{M}, \mathrm{Alg}^C}} \left[ \sum_{t=1}^{T} \mathrm{D_H}^2 (\overline{\mathrm{Alg}^E}(\cdot | \mathcal{T}_{t-1}, s), \mathrm{Alg}_{\widehat{\theta}}(\cdot | \mathcal{T}_{t-1}, s)) \right] \leq m_c \left[ c \cdot \frac{T \log(\delta^{-1} \mathcal{N}(1/(n_{\mathrm{pre}} T)^2) T)}{n_{\mathrm{pre}}} + T \epsilon_{\mathrm{real}} \right],$$

where $\epsilon_{\mathrm{real}}$ is defined in Assumption 5.2. Then we have $\mathbb{P}(\mathcal{E}) \geq 1 - 1/m_c - \delta$.

*Proof.* By Markov inequality, we have that with probability at most $1/m_c$,

$$\mathbb{E}_{\mathcal{T} \sim \mathbb{P}_{T-1}^{\mathbf{M}, \mathrm{Alg}^C}} \left[ \sum_{t=1}^{T} \mathrm{D_H}^2 (\overline{\mathrm{Alg}^E}(\cdot | \mathcal{T}_{t-1}, s), \mathrm{Alg}_{\widehat{\theta}}(\cdot | \mathcal{T}_{t-1}, s)) \right]$$

$$\geq m_c \cdot \mathbb{E}_{\mathbf{M} \sim \mathbb{P}^{\mathbf{M}}, s \sim \mathbb{P}_0^{\mathbf{M}}, \mathcal{T} \sim \mathbb{P}_{T-1}^{\mathbf{M}, \mathrm{Alg}^C}} \left[ \sum_{t=1}^{T} \mathrm{D_H}^2 (\overline{\mathrm{Alg}^E}(\cdot | \mathcal{T}_{t-1}, s), \mathrm{Alg}_{\widehat{\theta}}(\cdot | \mathcal{T}_{t-1}, s)) \right].$$

Meanwhile, by Lemma A.1, we know that with probability at most $\delta$, we have

$$\mathbb{E}_{\mathbf{M} \sim \mathbb{P}^{\mathbf{M}}, s \sim \mathbb{P}_0^{\mathbf{M}}, \mathcal{T} \sim \mathbb{P}_{T-1}^{\mathbf{M}, \mathrm{Alg}^C}} \left[ \sum_{t=1}^{T} \mathrm{D_H}^2 (\overline{\mathrm{Alg}^E}(\cdot | \mathcal{T}_{t-1}, s), \mathrm{Alg}_{\widehat{\theta}}(\cdot | \mathcal{T}_{t-1}, s)) \right]$$

$$\geq c \cdot \frac{T \log(\delta^{-1} \cdot \mathcal{N}(1/(n_{\mathrm{pre}} T)^2) T)}{n_{\mathrm{pre}}} + T \epsilon_{\mathrm{real}}.$$

Therefore, by the union bound, we have $\mathbb{P}(\mathcal{E}) \geq 1 - \delta - 1/m_c$. $\qquad \square$

Now we begin to prove Theorem 5.4.

*Proof.* We following the proof steps in Lin et al. (2023). We suppose that the event $\mathcal{E}$ denoted in Lemma A.2 holds. We first bound the difference of reward by the difference between their distribution distance. Let $\widehat{\mathbb{P}}$ be an arbitrary distribution over $\mathcal{T}$. Then we have

$$\mathbb{E}_{\mathcal{T} \sim \widehat{\mathbb{P}}(\cdot)} |\mathbb{E}_{a \sim \overline{\mathrm{Alg}^E}(\cdot | \mathcal{T}, s)} r(s, a) - \mathbb{E}_{a \sim \mathrm{Alg}_{\widehat{\theta}}(\cdot | \mathcal{T}, s)} r(s, a)|$$

$$\leq \mathbb{E}_{\mathcal{T} \sim \widehat{\mathbb{P}}(\cdot)} \mathrm{D_{TV}} (\overline{\mathrm{Alg}^E}(\cdot | \mathcal{T}, s), \mathrm{Alg}_{\widehat{\theta}}(\cdot | \mathcal{T}, s))$$

$$\leq \mathbb{E}_{\mathcal{T} \sim \widehat{\mathbb{P}}(\cdot)} \mathrm{D_H} (\overline{\mathrm{Alg}^E}(\cdot | \mathcal{T}, s), \mathrm{Alg}_{\widehat{\theta}}(\cdot | \mathcal{T}, s)), \tag{A.1}$$

the first inequality holds due to the fact $|r| \leq 1$ and the property of TV distance, the second one holds since $\mathrm{D_{TV}} \leq \mathrm{D_H}$. Starting from (A.1) we have

$$\mathbb{E}_{\mathcal{T} \sim \widehat{\mathbb{P}}} \mathrm{D_H} (\overline{\mathrm{Alg}^E}(\cdot | \mathcal{T}, s), \mathrm{Alg}_{\widehat{\theta}}(\cdot | \mathcal{T}, s))$$

$$= \mathbb{E}_{\mathcal{T} \sim \mathbb{P}_{T-1}^{\mathbf{M}, \mathrm{Alg}^C}} \mathrm{D_H} (\overline{\mathrm{Alg}^E}(\cdot | \mathcal{T}, s), \mathrm{Alg}_{\widehat{\theta}}(\cdot | \mathcal{T}, s)) \cdot \frac{\widehat{\mathbb{P}}(\mathcal{T})}{\mathbb{P}_{T-1}^{\mathbf{M}, \mathrm{Alg}^C}(\mathcal{T})}$$

$$\leq \sqrt{\underbrace{\mathbb{E}_{\mathcal{T}\sim\mathbb{P}^{\mathbf{M},\mathrm{Alg}^C}_{T-1}}\mathrm{D_H}^2(\overline{\mathrm{Alg}^E}(\cdot|\mathcal{T},s),\mathrm{Alg}_{\widehat{\boldsymbol{\theta}}}(\cdot|\mathcal{T},s))}_{I_1}} \cdot \sqrt{\underbrace{\mathbb{E}_{\mathcal{T}\sim\mathbb{P}^{\mathbf{M},\mathrm{Alg}^C}_{T-1}}\left(\frac{\widehat{\mathbb{P}}(\mathcal{T})}{\mathbb{P}^{\mathbf{M},\mathrm{Alg}^C}_{T-1}(\mathcal{T})}\right)^2}_{I_2}}, \quad (\mathrm{A.2})$$

where the first inequality holds due to Cauchy-Schwarz inequality. For $I_1$, we use Lemma A.1. Notice that the length of $|\mathcal{T}| = T - 1$ and the definition of $\epsilon_{\mathrm{pretrain}}$, we have

$$I_1 \leq (\epsilon_{\mathrm{pretrain}}/C_{\mathtt{Dec}})^2. \quad (\mathrm{A.3})$$

For $I_2$, by the definition of $\chi^2$ distance, we have

$$\begin{aligned}
I_2 &= \mathbb{E}_{\mathcal{T}\sim\widehat{\mathbb{P}}}\frac{\widehat{\mathbb{P}}(\mathcal{T})}{\mathbb{P}^{\mathbf{M},\mathrm{Alg}^C}_{T-1}(\mathcal{T})} \\
&= \mathbb{E}_{\mathcal{T}\sim\widehat{\mathbb{P}}}\frac{\widehat{\mathbb{P}}(\mathcal{T})}{\mathtt{Dec}_{T-1}(\mathcal{T}|s)} \cdot \frac{\mathtt{Dec}_{T-1}(\mathcal{T}|s)}{\mathbb{P}^{\mathbf{M},\mathrm{Alg}^C}_{T-1}(\mathcal{T})} \\
&\leq C^2_{\mathtt{Dec}}[1 + \chi^2(\widehat{\mathbb{P}}(\cdot),\mathtt{Dec}_{T-1}(\cdot|s))]. \quad (\mathrm{A.4})
\end{aligned}$$

where the inequality holds due to Assumption 5.3. Substituting (A.3) and (A.4) into (A.2), and substituting (A.2) into (A.1), we have

$$|\mathbb{E}_{\mathcal{T}\sim\widehat{\mathbb{P}},a\sim\overline{\mathrm{Alg}^E}(\cdot|\mathcal{T},s)}r(s,a) - \mathbb{E}_{\mathcal{T}\sim\widehat{\mathbb{P}},a\sim\mathrm{Alg}_{\widehat{\boldsymbol{\theta}}}(\cdot|\mathcal{T},s)}r(s,a)|$$

$$\leq \epsilon_{\mathrm{pretrain}}\sqrt{1 + \chi^2(\widehat{\mathbb{P}}(\cdot),\mathtt{Dec}_{T-1}(\cdot|s))}, \quad (\mathrm{A.5})$$

holds for any $\widehat{\mathbb{P}} \in \mathcal{P}$. Finally, we have

$$\mathbb{E}_{\mathcal{T}^s\sim\widehat{\mathbb{P}}^s(\cdot|\mathcal{D},s),a\sim\overline{\mathrm{Alg}^E}(\cdot|\mathcal{T}^s,s)}r(s,a)$$

$$\geq \mathbb{E}_{\mathcal{T}^s\sim\widehat{\mathbb{P}}^s(\cdot|\mathcal{D},s),a\sim\mathrm{Alg}_{\widehat{\boldsymbol{\theta}}}(\cdot|\mathcal{T}^s,s)}r(s,a) - \epsilon_{\mathrm{pretrain}}\sqrt{1 + \chi^2(\widehat{\mathbb{P}}^s(\cdot|\mathcal{D},s),\mathtt{Dec}_{T-1}(\mathcal{T}|s))}$$

$$\geq \mathbb{E}_{\mathcal{T}^s\sim\mathbb{P}^{*,s}(\cdot|\mathcal{D},s),a\sim\mathrm{Alg}_{\widehat{\boldsymbol{\theta}}}(\cdot|\mathcal{T}^s,s)}r(s,a) - \epsilon_{\mathrm{pretrain}}\sqrt{1 + \chi^2(\mathbb{P}^{*,s}(\cdot|\mathcal{D},s),\mathtt{Dec}_{T-1}(\mathcal{T}|s))},$$

$$\geq \mathbb{E}_{\mathcal{T}^s\sim\mathbb{P}^{*,s}(\cdot|\mathcal{D},s),a\sim\overline{\mathrm{Alg}^E}(\cdot|\mathcal{T}^s,s)}r(s,a) - 2\epsilon_{\mathrm{pretrain}}\sqrt{1 + \chi^2(\mathbb{P}^{*,s}(\cdot|\mathcal{D},s),\mathtt{Dec}_{T-1}(\mathcal{T}|s))},$$

$$\geq \mathbb{E}_{\mathcal{T}^s\sim\mathbb{P}^{*,s}(\cdot|\mathcal{D},s),a\sim\overline{\mathrm{Alg}^E}(\cdot|\mathcal{T}^s,s)}r(s,a) - 2\epsilon_{\mathrm{pretrain}}C_{\mathtt{Dec}}\sqrt{1 + \chi^2(\mathbb{P}^{*,s}(\cdot|\mathcal{D},s),\mathbb{P}^{\mathbf{M},\mathrm{Alg}^C}_{T-1}(\cdot))},$$

where the first inequality holds due to (A.5), the second one holds due to the selection rule of $\widehat{\mathbb{P}}^s$, the third one holds due to (A.5) and the last one holds due to Assumption 5.3. This concludes our proof.

$\square$

## A.2 ONLINE ALGORITHM

We also consider an analysis for a variant of OFFLINECOPS to the online setting. Here, let $\mathcal{P} = \{\mathbb{P}^1(\cdot|\cdot,\cdot),\ldots,\mathbb{P}^{|\mathcal{P}|}(\cdot|\cdot,\cdot)\} \subseteq 2^{\mathrm{T}_{t-1}\times\mathrm{S}\to\Delta(\mathrm{T}_{t-1})}$ which includes mappings that map an experience collection $\mathcal{T}_{t-1}$ and a test state $s$ to a distribution over $\mathrm{T}_{t-1}$. Each $\mathbb{P}^i$ can be thought as a strategy to pick the experience collection that depends on the past observations. At step $t$, we have history $\mathcal{H}_{t-1} = \{s_1,a_1,r_1,\ldots,s_{t-1},a_{t-1},r_{t-1}\}$. Then the agent receives $s_t \sim \mathbb{P}^{\mathbf{M}_t}_0$, where $\mathbf{M}_t \sim \mathbb{P}^{\mathbf{M}}$. Then the agent selects $\mathbb{P}_t$ by some algorithm and samples $\mathcal{T}_{t-1} \sim \mathbb{P}_t(\cdot|\mathcal{H}_{t-1},s_t)$. Then the agent takes the action $a_t \sim \mathrm{Alg}_{\widehat{\boldsymbol{\theta}}}(\cdot|\mathcal{T}_{t-1},s_t)$. Her goal is to minimize the following regret:

$$\mathrm{Regret}_T := \sum_{t=1}^{T}\mathbb{E}_{\mathbf{M}_t\sim\mathbb{P}^{\mathbf{M}},s_t\sim\mathbb{P}^{\mathbf{M}_t}_0}\left[\max_{\substack{\mathbb{P}^i\in\mathcal{P}\\\bar{a}\sim\overline{\mathrm{Alg}^E}(\cdot|\mathcal{T}_{t-1},s_t)}}\mathbb{E}_{\mathcal{T}_{t-1}\sim\mathbb{P}^i(\cdot|\mathcal{H}_{t-1}),}r(s_t,\bar{a}) - \mathbb{E}_{\substack{\mathcal{T}_{t-1}\sim\mathbb{P}_t(\cdot|\mathcal{H}_{t-1}),\\a_t\sim\mathrm{Alg}^E(\cdot|\mathcal{T}_{t-1},s_t)}}r(s_t,a_t)\right]. \quad (\mathrm{A.6})$$

We propose the algorithm ONLINECOPS in Algorithm 3. Similar to OFFLINECOPS, ONLINECOPS adapts an decoder that takes the current state as its input and outputs a distribution of the experience

collection $\mathcal{T}$, which aims to estimate the LLM output distribution $\mathbb{P}_{t-1}^{\mathbf{M}_t,\mathrm{Alg}^C}$. Unlike OFFLINECOPS, the optimization goal of ONLINECOPS in (A.7) is similar to the *optimistic principle* that originates from the online decision-making problems (Abbasi-Yadkori et al., 2011), which aims to maximize both the reward and the distribution distance between the decoder distribution $\mathrm{Dec}_{t-1}$ and the selected one $\widehat{\mathbb{P}}^t$. Meanwhile, note that the selected experience collection distribution only depends on the past history $\mathcal{H}_{t-1}$, which is small in the early stage of the online decision-making process. We have the following theorem to demonstrate the theoretical guarantee of ONLINECOPS.

---

**Algorithm 3** ONLINECOPS

---

**Require:** LLM $\mathrm{Alg}_{\widehat{\boldsymbol{\theta}}}(\cdot|\cdot,\cdot)$, candidate experience collection distribution $\mathcal{P}$, pretraining error parameter $\epsilon_{\mathrm{pretrain}}$, task decoder $\mathrm{Dec}$.
1: Let $\mathcal{H}_0 = \emptyset$.
2: **for** $t = 1, \ldots, T$ **do**
3:  Generate $\mathbf{M}_t \sim \mathbb{P}^{\mathtt{M}}$, receive $s_t \sim \mathbb{P}_0^{\mathbf{M}_t}$, decode $\mathrm{Dec}_{t-1}(\cdot|s_t)$
4:  Select $\widehat{\mathbb{P}}^t$ from $\mathcal{P}$ that maximizes the following:

$$\widehat{\mathbb{P}}^t = \operatorname*{argmax}_{\substack{\widehat{\mathbb{P}} \in \mathcal{P}}} \mathbb{E}_{\substack{\mathcal{T} \sim \widehat{\mathbb{P}}(\cdot|\mathcal{H}_{t-1}, s_t), \\ a \sim \mathrm{Alg}_{\widehat{\boldsymbol{\theta}}}(\cdot|\mathcal{T}, s_t)}} r(s_t, a) + \epsilon_{\mathrm{pretrain}} \sqrt{1 + \chi^2(\widehat{\mathbb{P}}(\cdot|\mathcal{H}_{t-1}, s_t), \mathrm{Dec}_{t-1}(\cdot|s_t))}. \quad (\mathrm{A.7})$$

5:  Generate $\mathcal{T} \sim \widehat{\mathbb{P}}^t(\cdot|\mathcal{H}_{t-1}, s_t)$ and obtain $a_t \sim \mathrm{Alg}_{\widehat{\boldsymbol{\theta}}}(\cdot|\mathcal{T}, s_t)$ and $r_t = r(s_t, a_t)$, set $\mathcal{H}_t = \mathcal{H}_{t-1} \cup (s_t, a_t, r_t)$.
6: **end for**

---

**Theorem A.3.** By setting

$$\epsilon_{\mathrm{pretrain}} = C_{\mathrm{Dec}} \cdot T^2 \cdot \sqrt{5 \cdot \frac{T \log(\mathcal{N}(1/(n_{\mathrm{pre}}T)^2)T^2)}{n_{\mathrm{pre}}} + T\epsilon_{\mathrm{real}}},$$

and denote $\mathbb{P}^{*,t} = \operatorname{argmax}_{\widehat{\mathbb{P}} \in \mathcal{P}} \mathbb{E}_{\substack{\mathcal{T}_{t-1} \sim \widehat{\mathbb{P}}(\cdot|\mathcal{H}_{t-1}, s_t), \\ \bar{a} \sim \overline{\mathrm{Alg}^E}(\cdot|\mathcal{T}_{t-1}, s_t)}} r(s_t, \bar{a})$, we have the following bound holds with probability at least $1 - 2/T$:

$$\mathrm{Regret}_T \leq 2 C_{\mathrm{Dec}} \epsilon_{\mathrm{pretrain}} \sum_{t=1}^{T} \sqrt{1 + \chi^2(\mathbb{P}^{*,t}(\cdot|\mathcal{H}_{t-1}, s_t), \mathbb{P}_{t-1}^{\mathbf{M}_t, \mathrm{Alg}^C}(\cdot))}.$$

*Proof.* Suppose we are at step $t$ and we condition on all past history $\mathcal{H}_{t-1} = (s_1, a_1, r_1, \ldots, s_{t-1}, a_{t-1}, r_{t-1})$.

Let $\mathbf{M}_t$ be the task at $t$ step and $s_t$ be the state observed. Then with probability at least $1 - 1/m_c - \delta$, the following event $\mathcal{E}_t$ holds:

$$\mathbb{E}_{\mathcal{T} \sim \mathbb{P}_{t-1}^{\mathbf{M}_t, \mathrm{Alg}^C}} \left[ \mathrm{D}_{\mathrm{H}}{}^2 \left( \overline{\mathrm{Alg}^E}(\cdot|\mathcal{T}_{t-1}, s_t), \mathrm{Alg}_{\widehat{\boldsymbol{\theta}}}(\cdot|\mathcal{T}_{t-1}, s_t) \right) \right]$$
$$\leq m_c \left[ c \cdot \frac{T \log(\delta^{-1} \mathcal{N}(1/(n_{\mathrm{pre}}T)^2)T^2)}{n_{\mathrm{pre}}} + T\epsilon_{\mathrm{real}} \right],$$

Now following (A.2) in the proof of Theorem 5.4, we still have

$$\mathbb{E}_{\mathcal{T} \sim \widehat{\mathbb{P}}} |\mathbb{E}_{a \sim \overline{\mathrm{Alg}^E}(\cdot|\mathcal{T}, s_t)} r(s, a) - \mathbb{E}_{a \sim \mathrm{Alg}_{\widehat{\boldsymbol{\theta}}}(\cdot|\mathcal{T}, s_t)} r(s, a)|$$

$$\leq \sqrt{\underbrace{\mathbb{E}_{\mathcal{T} \sim \mathbb{P}_{t-1}^{\mathbf{M}_t, \mathrm{Alg}^C}} \mathrm{D}_{\mathrm{H}}{}^2 \left( \overline{\mathrm{Alg}^E}(\cdot|\mathcal{T}, s_t), \mathrm{Alg}_{\widehat{\boldsymbol{\theta}}}(\cdot|\mathcal{T}, s_t) \right)}_{I_1} \cdot \underbrace{\mathbb{E}_{\mathcal{T} \sim \mathbb{P}_{t-1}^{\mathbf{M}_t, \mathrm{Alg}^C}} \left( \frac{\widehat{\mathbb{P}}(\mathcal{T})}{\mathbb{P}_{t-1}^{\mathbf{M}_t, \mathrm{Alg}^C}(\mathcal{T})} \right)^2}_{I_2}} \quad (\mathrm{A.8})$$

Then following Lemma A.2, under event $\mathcal{E}_t$, we have

$$I_1 \leq (\epsilon_{\mathrm{pretrain}}/C_{\mathrm{Dec}})^2, \quad \epsilon_{\mathrm{pretrain}}/C_{\mathrm{Dec}} = T^2 \cdot \sqrt{c \cdot \frac{T \log(\mathcal{N}(1/(n_{\mathrm{pre}}T)^2)T^2)}{n_{\mathrm{pre}}} + T\epsilon_{\mathrm{real}}}.$$

For $I_2$, similar to (A.4), we have

$$I_2 \leq C_{\texttt{Dec}}^2 [1 + \chi^2(\widehat{\mathbb{P}}(\cdot), \texttt{Dec}_{t-1}(\cdot|s_t))].$$

Therefore, we have for any $\widehat{\mathbb{P}}$,

$$|\mathbb{E}_{\mathcal{T}\sim\widehat{\mathbb{P}}(\cdot|\mathcal{H}_{t-1},s_t), a\sim\overline{\text{Alg}^E}(\cdot|\mathcal{T},s_t)} r(s_t,a) - \mathbb{E}_{\mathcal{T}\sim\widehat{\mathbb{P}}(\cdot|\mathcal{H}_{t-1},s_t), a\sim\text{Alg}_{\widehat{\theta}}(\cdot|\mathcal{T},s_t)} r(s_t,a)|$$

$$\leq \epsilon_{\text{pretrain}} \sqrt{1 + \chi^2(\widehat{\mathbb{P}}(\cdot|\mathcal{H}_{t-1},s_t), \texttt{Dec}_{t-1}(\cdot|s_t))}. \quad (A.9)$$

Taking union bound and let $m_c = T^2, \delta = 1/T^2$, then we get $\mathcal{E}_1, ..., \mathcal{E}_T$ hold with probability at least $1 - 2/T$. Next we bound the suboptimal gap at $t$ step as follows:

$$\mathbb{E}_{\mathcal{T}^{t-1}\sim\widehat{\mathbb{P}}^t(\cdot|\mathcal{H}_{t-1},s_t), a\sim\overline{\text{Alg}^E}(\cdot|\mathcal{T}^{t-1},s_t)} r(s_t,a)$$

$$\leq \mathbb{E}_{\substack{\mathcal{T}^{t-1}\sim\widehat{\mathbb{P}}^t(\cdot|\mathcal{H}_{t-1},s_t), \\ a\sim\text{Alg}_{\widehat{\theta}}(\cdot|\mathcal{T}^{t-1},s_t)}} r(s_t,a) + \epsilon_{\text{pretrain}} \sqrt{1 + \chi^2(\widehat{\mathbb{P}}^t(\cdot|\mathcal{H}_{t-1},s_t), \texttt{Dec}_{t-1}(\cdot|s_t))}$$

$$\leq \mathbb{E}_{\substack{\mathcal{T}^{t-1}\sim\mathbb{P}^{*,t}(\cdot|\mathcal{H}_{t-1},s_t), \\ a\sim\text{Alg}_{\widehat{\theta}}(\cdot|\mathcal{T}^{t-1},s_t)}} r(s_t,a) + \epsilon_{\text{pretrain}} \sqrt{1 + \chi^2(\mathbb{P}^{*,t}(\cdot|\mathcal{H}_{t-1},s_t), \texttt{Dec}_{t-1}(\cdot|s_t))}$$

$$\leq \mathbb{E}_{\substack{\mathcal{T}^{t-1}\sim\mathbb{P}^{*,t}(\cdot|\mathcal{H}_{t-1},s_t), \\ a\sim\overline{\text{Alg}^E}(\cdot|\mathcal{T}^{t-1},s_t)}} r(s_t,a) + 2\epsilon_{\text{pretrain}} \sqrt{1 + \chi^2(\mathbb{P}^{*,t}(\cdot|\mathcal{H}_{t-1},s_t), \texttt{Dec}_{t-1}(\cdot|s_t))}$$

$$\leq \mathbb{E}_{\substack{\mathcal{T}^{t-1}\sim\mathbb{P}^{*,t}(\cdot|\mathcal{H}_{t-1},s_t), \\ a\sim\overline{\text{Alg}^E}(\cdot|\mathcal{T}^{t-1},s_t)}} r(s_t,a) + 2C_{\texttt{Dec}}\epsilon_{\text{pretrain}} \sqrt{1 + \chi^2(\mathbb{P}^{*,t}(\cdot|\mathcal{H}_{t-1},s_t), \mathbb{P}_{t-1}^{\mathbf{M}_t,\text{Alg}^C}(\cdot))}, \quad (A.10)$$

where the first inequality holds due to (A.9), the second one holds due to the optimism principle, the third one holds due to (A.9), and the last one holds due to Assumption 5.3. Taking summation of (A.10) from 1 to $T$ concludes our proof. $\square$

Similar to Theorem 5.4 for the offline setting, Theorem A.3 also shares the following insights.

- The regret is controlled by the difference between the best experience collection generated distribution $\mathbb{P}^{*,t}$ and the experience collection distribution induced by the contextual algorithm at $t$-th step. Therefore, the best strategy overall is to select trajectories from the history $\mathcal{H}_{t-1}$ that can approximates the current task well to avoid the distribution shift.
- With a more powerful LLM, the $\epsilon_{\text{pretrain}}$ will be smaller, which means the selected experience collection can approximatethe best selection better.

## B ABLATION STUDY

In this section, we analyze how two key hyperparameters affect the performance of COPS: the scaling factor $c$ in Equation (2.3) and the number of in-context experiences $k$ placed at the beginning of prompts. We conducted experiments on the Alfworld benchmark using both Llama 3.1 8b and Llama 3.1 70b models.

For the scaling factor $c$, we tested four settings: $c = 0$, 1, 5 and 10, while keeping the number of in-context experiences fixed at $k = 5$ (see Figures 3(a) and 3(b)). Our findings indicate that for smaller models like Llama 3.1 8b, a small but non-zero value of $c$ (e.g., $c = 1$) generally yields better performance (Figure 3(a)). This suggests that moderate scaling effectively balances model adaptability and robustness on less capable models.

Regarding the number of in-context experiences $k$, we evaluated values ranging from 1 to 10, setting $c = 0$ (see Figures 3(c) and 3(d)). We observed that performance improves as $k$ increases up to $k = 3$, after which it plateaus for both model sizes. This result indicates that while increasing the in-context experience size enhances performance to a point, adding more than three experiences may not offer substantial gains.

Our ablation study reveals that tuning key hyperparameters in COPS is crucial for optimal performance. Specifically, for smaller models, a small but non-zero scaling factor $c$ (e.g., $c = 1$) effectively

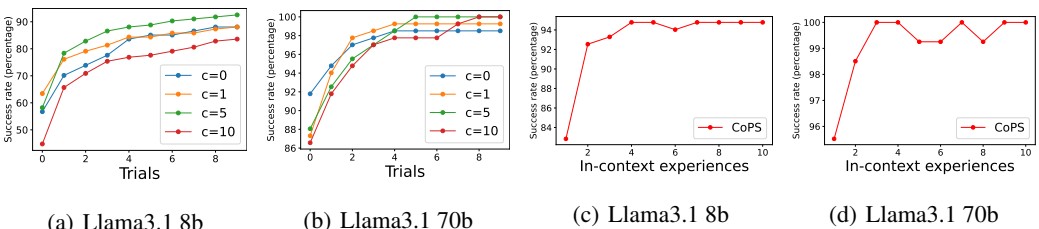

(a) Llama3.1 8b        (b) Llama3.1 70b        (c) Llama3.1 8b        (d) Llama3.1 70b

Figure 3: Performance impact of hyperparameters $c$ (scaling factor) and $k$ (number of in-context experiences) on the Alfworld benchmark for both Llama 3.1 8b and Llama 3.1 70b models.

balances adaptability and robustness. Additionally, increasing the number of in-context experiences $k$ enhances performance up to $k = 3$, beyond which additional experiences offer minimal gains. These insights provide practical guidance for hyperparameter selection, ensuring that CoPS can be efficiently deployed across various settings to maximize its sequential reasoning capabilities.

## C   MORE EXPERIMENT DETAILS

In this section, we provide additional details on our experiments in Section 4. The tables included below outline the token counts and hyperparameter settings that were used throughout the evaluation process.

Table 4: Token generation count for each of the **Webshop** experiments. It's worth noticing that for each model the LATS token generation count is at least 5 times to CoPS.

| Algorithm | Reflexion | RAP | LATS | CoPS |
|-----------|-----------|-----|------|------|
| **Llama3.1 8b** | 159131 | 107504 | 1555365 | 314336 |
| **Llama3.1 70b** | 125406 | 109245 | 1058752 | 113849 |

Table 5: Hyperparameter settings ($k$ and $c$) for different benchmarks and model sizes.

| Benchmark | Alfworld | Webshop | HotPotQA |
|-----------|----------|---------|----------|
| **Llama3.1 8b** | $k = 5, c = 5$ | $k = 5, c = 0$ | $k = 5, c = 5$ |
| **Llama3.1 70b** | $k = 5, c = 5$ | $k = 5, c = 0$ | $k = 5, c = 0$ |

## D   QUALITY OF DEMONSTRATIONS

In our realistic implementation of CoPS, we only utilized successful tractors following other related works. However, in our theoretical analysis, we use the measurement we designed in equation 2.2, which considers both the successful and failed trajectories and calculates the similarity between the experience and our current task. However, in realistic implementation, the trajectories that gain high similarity scores are successful, thus we only utilize successful trajectories due to limited compute budgets.

This brings concerns about the impact of suboptimal demonstrations, for which, we conducted an ablation study on the Alfworld benchmark, comparing top-k and bottom-k successful trajectories ranked by the similarity score. The results are shown in Table 6.

Table 6: Performance comparison of using top-k and bottom-k successful trajectories as demonstrations on Alfworld benchmark using Llama3.1 8B Instruct.

| Retrieval Method | Performance (Success Rate %) |
|------------------|------------------------------|
| Top-5 | $93.6 \pm 1.0$ |
| Bottom-5 | $83.0 \pm 3.9$ |

These results in Table 6 demonstrate that the quality of retrieved demonstrations significantly affects performance, with top-k successful trajectories outperforming bottom-k successful trajectories by a substantial margin. This underscores the importance of selecting high-quality trajectories.

## E    REPEATED EXPERIMENTS

To demonstrate the robustness of CoPS, we use multiple seeds to run CoPS on all three benchmarks. The repeated experiment results are shown in Table 7.

| Benchmark | Model | Mean | Std |
|-----------|-------|------|-----|
| HotpotQA | 8B | 53.6 | 1.5 |
| | 70B | 62.8 | 1.3 |
| Webshop | 8B | 47.2 | 1.6 |
| | 70B | 51.2 | 2.7 |
| Alfworld | 8B | 93.6 | 1.0 |
| | 70B | 100.0 | 0.0 |

Table 7: Mean and standard deviation results of LLaMA 3.1 Instruct model on three benchmarks.

The results presented in Table 7 demonstrate the robustness of CoPS across multiple runs and benchmarks. For example, on the Alfworld benchmark with the LLaMA 3.1 70B model, our method consistently achieved perfect scores ($100.0 \pm 0.0$), highlighting its stability. On other benchmarks such as HotpotQA and Webshop, the relatively low standard deviations further validate the consistency of our approach, even under varied experimental conditions. These findings underscore the reliability and robustness of CoPS, reinforcing its applicability to diverse real-world scenarios.

## F    PERFORMANCE ON CLOSE-SOURCED MODELS

We add additional experiments to evaluate the performance of CoPS based on SOTA close-sourced GPT and Claude models. The detailed performance is shown in Table 8. From the results, we find that CoPS works well with these close-sourced models and achieves reasonably high performance compared with open-source models.

Table 8: Performance comparison of CoPS on different benchmarks using GPT and Claude family models.

| Model | Alfworld | Webshop | HotPotQA |
|-------|----------|---------|----------|
| GPT-4o | 100 | 56 | 67 |
| Claude 3.5-Sonnet | 100 | 58 | 66 |

## G    IMPACT OF CROSS-TASK EXPERIENCES

CoPS utilized cross-task experiences to boost the performance of LLM agents. This brings concerns about whether CoPS can achieve similar performance just using the experiences from the failed trajectories of the same task. While leveraging single-task experience might seem ideal, practical scenarios often necessitate relying on experiences from relevant but distinct tasks, which introduces additional challenges. To address this concern, we conducted an ablation study comparing the performance of CoPS with and without cross-task experience on the Alfworld benchmark. The results are shown in Table 9.

Table 9: Impact of cross-task experience on performance (success rate) using the Alfworld benchmark with Llama 3.1 8B Instruct.

| Method | Success Rate % |
|--------|----------------|
| with cross-task experience | 94 |
| only same-task experience | 57 |

These results clearly demonstrate the significant contribution of cross-task experience to performance improvement, with a nearly twofold increase in success rate compared to using only same-task experience.

## H    IMPACT OF RETRIEVAL METHODS

In our main results, CoPS utilized semantic search (semantic embedding model and distance-based retrieval) to retrieve cross-task experiences. This raises concerns about the impact of the retrieval methods. To evaluate the impact of the memory retrieval method, we conduct an ablation study on the AlfWorld benchmark with the llama 3.1 8b Instruct model. The results are summarized in Table 10.

Table 10: Performance comparison of different experiences retrieval strategies of CoPS on Alf-World benchmark. The experiments are repeated 5 times and reported in mean + std style.

| Retrieval Method | Success Rate % |
|---|---|
| Semantic Search (embedding model) | $93.6 \pm 1.0$ |
| Keyword-Based (BM25) | $94.1 \pm 1.2$ |
| Hybrid (BM25 + Short Summarization Embedding) | $91.3 \pm 1.4$ |

These results indicate that semantic search and keyword-based approaches perform comparably well, whereas the hybrid approach shows a slight performance drop, potentially due to the added complexity of combining methods.

## I    IMPACT ON MEMORY SIZE

In our initial experiments, we assumed a sufficiently large memory bank and did not model forgetting, which ignored the importance of memory managing and forgetting mechanisms, especially for long-term agent deployment. To address this concern, we conducted a new ablation study on the Alf-world benchmark with varying memory sizes to evaluate the system's robustness under constrained memory conditions.

In our main results, we retained all trajectories from different trials and conducted experience retrieval across the entire memory bank. For this ablation, we introduced a fixed memory size and implemented a dynamic forgetting mechanism, where low-scored experiences were discarded once the memory capacity was reached. The results are summarized in Table 11.

Table 11: Performance evaluation under constrained memory sizes on AlfWorld benchmark. Note that our main result sets memory size to 10, as it corresponds to 10 trials. Therefore, our main results do not discard any experiences. In our ablation study, we run CoPS for 50 trials, thus for sizes 5 and 10 in the ablation study, experiences were dynamically discarded when the memory limit was exceeded.

| Memory Size | Performance (Success Rate %) |
|---|---|
| 50 | $95.6 \pm 2.7$ |
| 10 | $94.0 \pm 1.1$ |
| 5 | $87.2 \pm 0.8$ |

The results demonstrate that CoPS maintains robust performance even with constrained memory sizes, with only a slight drop in success rate when the memory size is reduced from 50 to 10. This indicates that our experience selection strategy is effective even under memory limitations. However, as expected, significant reductions in memory size (e.g., to 5) lead to performance degradation due to more aggressive forgetting of potentially useful experiences. These findings highlight the adaptability of CoPS to resource-constrained scenarios while also emphasizing the trade-offs introduced by limited memory capacity.

## J    LIMITATIONS

While CoPS shows clear improvements over existing methods, it has several limitations. Its effectiveness heavily depends on the quality and diversity of the experiences in the memory bank, mean-

ing that outdated or poorly aligned experiences can reduce its performance. Additionally, CoPS is sensitive to hyperparameters like the scaling factor and the number of in-context experiences, which may require time-consuming tuning that doesn't always generalize well across different tasks or models. Finally, the theoretical guarantees we provide also rely on assumptions about the accuracy of the decoder and specific pre-training properties of the LLM, which may not always hold in real-world scenarios.

## K    FUTURE WORK

Looking ahead, several research directions could further improve CoPS.[4] These include developing adaptive hyperparameter tuning methods, exploring dynamic memory management to keep experiences relevant, and adding ways to assess experience quality. Additionally, adapting CoPS for multi-agent systems and integrating it with reinforcement learning could enable collaborative learning and continuous performance improvement. Pursuing these areas will help LLM agents handle even more complex sequential reasoning tasks.

## L    PROMPT TEMPLATE

Our prompt framework is designed with the considerations of simplicity, efficiency, and generalizability. Moreover, we aim to leverage the high-performing long-context capabilities of modern LLMs ((Dubey et al., 2024)) to a maximal degree. In one sentence to describe our prompting philosophy: previous success trajectories are seen by the agent as *experiences* it has gone through in the same trial. As an example from the Alfworld experiment, a in-context demonstration is insert into the prefix of the agent's prompt as follows:

---

[User] Interact with a household to solve a task. Here are a few examples.
You are in the middle of a room. Looking quickly around you, you see a cabinet 13, a cabinet 12, a cabinet 11, a cabinet 10, a cabinet 9, a cabinet 8, a cabinet 7, a cabinet 6, a cabinet 5, a cabinet 4, a cabinet 3, a cabinet 2, a cabinet 1, a coffeemachine 1, a countertop 1, a diningtable 1, a drawer 1, a fridge 1, a garbagecan 1, a microwave 1, a shelf 3, a shelf 2, a shelf 1, a sinkbasin 1, a stoveburner 4, a stoveburner 3, a stoveburner 2, a stoveburner 1, and a toaster 1.
Your task is to: put a hot apple in fridge.

[Assistant] think: To solve the task, I need to find and take an apple, then heat it with microwave, then put it in fridge.

[User] OK.

[Assistant] think: First I need to find an apple. An apple is more likely to appear in fridge (1), diningtable (1), coffeetable (1), drawer (1), cabinet (1-13), garbagecan (1). I can check one by one, starting with fridge 1.

[User] OK.

[Assistant] go to fridge 1

[User] The fridge 1 is closed.

**[experience detail omitted]**

[Assistant] go to fridge 1

[User] The fridge 1 is open. In it, you see a cup 1, and a egg 1.

[Assistant] put apple 1 in/on fridge 1

[User] You put the apple 1 in/on the fridge 1.

---

**[more in-context experiences]**

[User] Examples finished. Here is the task:
You are in the middle of a room. Looking quickly around you, you see a cabinet 6, a cabinet 5, a cabinet 4, a cabinet 3, a cabinet 2, a cabinet 1, a coffeemachine 1, a countertop 3, a countertop 2, a countertop 1, a drawer 3, a drawer 2, a drawer 1, a fridge 1, a garbagecan 1, a microwave 1, a shelf 3, a shelf 2, a shelf 1, a sinkbasin 1, a stoveburner 4, a stoveburner 3, a stoveburner 2, a stoveburner 1, and a toaster 1.
Your task is to: put a saltshaker in drawer.

[Assistant] think: To solve the task, I need to find and take a saltshaker, then put it in/on a drawer.

**[real trial begins]**

