# OpenReview forum: "CoPS: Empowering LLM Agents with Provable Cross-Task Experience Sharing"
_ICLR.cc/2025/Conference — Submitted to ICLR 2025_

### Official Review · Reviewer_ct7E · 2024-11-04

**Soundness:** 3
**Presentation:** 2
**Contribution:** 2
**Rating:** 5
**Confidence:** 3

**Summary:**

This paper proposes CoPS (Cross-Task Experience Sharing), an algorithm that aims to enhance LLM-based agents’ sequential reasoning by leveraging experiences across different tasks. The method uses a pessimism-based strategy to select relevant experiences from a memory bank while minimizing distribution shift risks. The authors evaluate CoPS on three benchmarks (Alfworld, Webshop, HotPotQA) and claim superior performance compared to baselines like Reflexion and RAP.

**Strengths:**

- The core idea of leveraging cross-task experiences for LLM agents is novel and potentially impactful. The pessimism-based selection strategy provides a theoretically grounded approach to experience sharing.
 - The implementation is relatively straightforward and generalizable across different environments, requiring minimal task-specific modifications.
 - The empirical results show promising performance improvements, particularly with smaller models like Llama 3.1 8b, suggesting potential resource efficiency benefits.

**Weaknesses:**

1. The idea seems similar to the Retrieval-Augmented Generation (RAG) technique. The paper lacks a comparison and discussion with traditional RAG approaches that also leverage external knowledge for LLM enhancement. The pessimism-based selection strategy could be better positioned against existing RAG retrieval methods like hybrid search and recursive retrieval[1].
2. The relationship to LLM agent memory mechanisms is insufficiently explored. For instance, no comparison is made with memory bank approaches that handle both short-term and long-term memory [2,3,4]. The paper should discuss how CoPS differs from or improves upon existing memory management solutions in LLM agents.
3. The current implementation lacks consideration of hybrid memory architectures that combine both short-term and long-term memory components. The system could benefit from incorporating recent advances in memory management like episodic memory modules or hierarchical attention mechanisms. Also, the paper doesn't address how the experience selection strategy could be enhanced with modern RAG techniques like recursive retrieval or adaptive retrieval mechanisms.
4. No ablation studies comparing different memory retrieval strategies (e.g., semantic search vs. keyword-based vs. hybrid approaches). Missing evaluation of memory retention and recall over extended periods, which is crucial for long-term agent deployment. Limited analysis of how the system handles memory updates and forgetting mechanisms compared to other memory-augmented LLM approaches[3].
5. Presentation could be improved. For instance, Fig. 1 does not with adequate explanation and it is hard to understand what does the example task means.

Refs:
[1] Retrieval Augmented Generation (RAG) for LLMs https://www.promptingguide.ai/research/rag

[2] Zhang, Zeyu, et al. "A survey on the memory mechanism of large language model based agents." arXiv preprint arXiv:2404.13501 (2024).

[3] MemoryBank: Enhancing Large Language Models with Long-Term Memory, https://ojs.aaai.org/index.php/AAAI/article/view/29946

[4] Wang, Guanzhi, et al. "Voyager: An open-ended embodied agent with large language models." arXiv preprint arXiv:2305.16291 (2023).

[5] A Survey on Retrieval-Augmented Text Generation for Large Language Models - NASA ADS https://ui.adsabs.harvard.edu/abs/2024arXiv240410981H/abstract

**Questions:**

Apart from the weakness section, I also have a few questions:

1. How is the task defined? It is a bit unclear to me in the experiments -- does the author use the experience from the same benchmark or from other benchmarks as well?

2. How is the sampled experience number selected? How would the number affect the performance? Note that more sampled experience will result in more context length.

---

> ### Author Response · Authors · 2024-11-24
> **Reply to Reviewer ct7E (Comment 1)**
>
> # Reply to Reviewer ct7E
>
> We sincerely thank your thorough and insightful feedback, which has significantly contributed to improving the quality and clarity of our paper. We have carefully addressed all comments and concerns in our revised submission and provided detailed responses to each point. If there are any remaining questions or further points for clarification, please do not hesitate to raise them. We greatly value the opportunity to engage in further discussions and refine our work based on your suggestions. Thank you once again for your time and effort in reviewing our paper.
>
> ## Comment 1
>
> > *The idea seems similar to the Retrieval-Augmented Generation (RAG) technique. The paper lacks a comparison and discussion with traditional RAG approaches that also leverage external knowledge for LLM enhancement. The pessimism-based selection strategy could be better positioned against existing RAG retrieval methods like hybrid search and recursive retrieval[1].*
>
> ## Response
>
> Thank you for this thoughtful suggestion. We acknowledge that our method *CoPS* is conceptually related to Retrieval-Augmented Generation (RAG) and can be seen as a specialized adaptation of this paradigm. While traditional RAG techniques focus on retrieving information from external knowledge to enhance general-purpose language generation, *CoPS* uniquely retrieves an agent's own experiences, specifically tailored for decision-making and adaptation in LLM agent settings without expensive manual experiences/knowledge collections.
>
> This distinction allows our method to better address challenges inherent to LLM agents, such as leveraging past trajectories for task-specific improvements. Moreover, the incorporation of a pessimism-based selection strategy provides a principled approach to retrieval that ensures robust performance under uncertainty, setting it apart from traditional RAG techniques like hybrid search or recursive retrieval. As shown in our experimental results, this targeted adaptation results in superior sample efficiency and task performance, which are critical in real-world agent scenarios.
>
> **We've discussed these differences in detail in the Related Works and Theoretical Analysis sections of our paper and appreciate the opportunity to further clarify the unique contributions of our method.**
>
> ---

---

> ### Author Response · Authors · 2024-11-24
> **Reply to Reviewer ct7E (Comment 2)**
>
> ## Comment 2
>
> > *The relationship to LLM agent memory mechanisms is insufficiently explored. For instance, no comparison is made with memory bank approaches that handle both short-term and long-term memory [2,3,4]. The paper should discuss how CoPS differs from or improves upon existing memory management solutions in LLM agents. The current implementation lacks consideration of hybrid memory architectures that combine both short-term and long-term memory components. The system could benefit from incorporating recent advances in memory management like episodic memory modules or hierarchical attention mechanisms. Also, the paper doesn't address how the experience selection strategy could be enhanced with modern RAG techniques like recursive retrieval or adaptive retrieval mechanisms.*
>
> ## Response
>
> Thank you for these insightful comments and for highlighting the potential connections between *CoPS* and existing memory management mechanisms. We agree that exploring these relationships represents an important direction for future work and could further enhance the applicability of our framework.
>
> Nonetheless, the primary focus of *CoPS* is to facilitate cross-task experience sharing by leveraging the similarities between experiences, without explicitly incorporating or requiring long-term or short-term memory management into its current implementation. As such, our work focuses on designing a generalizable experience selection strategy rather than addressing memory organization or management in detail. Therefore, comparisons with memory bank approaches or hybrid memory architectures are beyond the scope of this paper. However, we recognize the significant opportunities for synergies between these areas.
>
> For instance, integrating *CoPS* with hybrid memory systems that combine short-term and long-term memory could improve its ability to retain and retrieve task-relevant experiences across extended interactions. Episodic memory modules, for example, could enable more fine-grained handling of temporally structured information, while hierarchical attention mechanisms could help prioritize the most relevant experiences for retrieval. These enhancements could strengthen the adaptability of *CoPS* in scenarios requiring continuous learning or interaction over prolonged periods.
>
> Additionally, incorporating modern RAG techniques such as recursive retrieval or adaptive retrieval mechanisms could further refine *CoPS*'s experience selection strategy. Recursive retrieval could improve the precision of memory queries by iteratively refining retrieval contexts, while adaptive mechanisms could dynamically adjust retrieval strategies based on the task or environment, enhancing both efficiency and effectiveness.
>
> **We acknowledge the value of these advanced memory management techniques and believe that their integration with CoPS has the potential to unlock new capabilities. We plan to explore these possibilities in future work and appreciate your suggestions, which provide a valuable perspective for extending the scope and impact of our research.**
>
> ---

---

> ### Author Response · Authors · 2024-11-24
> **Reply to Reviewer ct7E (Comment 3 ~ Comment 4)**
>
> ## Comment 3
>
> > *No ablation studies comparing different memory retrieval strategies (e.g., semantic search vs. keyword-based vs. hybrid approaches). Missing evaluation of memory retention and recall over extended periods, which is crucial for long-term agent deployment. Limited analysis of how the system handles memory updates and forgetting mechanisms compared to other memory-augmented LLM approaches[3].*
>
> ### Memory Retrieval Strategies
>
> Thank you for these valuable observations. We think our approach can be categorized into semantic search methods. To evaluate different memory retrieval strategies, we performed an ablation study on the AlfWorld benchmark with the LLaMA 3.1 8B Instruct model. The results are summarized below:
>
> | **Retrieval Method**                     | **Success Rate (%)**  |
> |------------------------------------------|------------------------|
> | **Semantic Search (embedding model)**    | 93.6 ± 1.0            |
> | **Keyword-Based (BM25)**                 | 94.1 ± 1.2            |
> | **Hybrid (BM25 + Short Summarization Embedding)** | 91.3 ± 1.4   |
>
> These results indicate that semantic search and keyword-based approaches perform comparably well, whereas the hybrid approach shows a slight performance drop, potentially due to the added complexity of combining methods. **We have included this ablation study in Appendix H "IMPACT OF RETRIEVAL METHODS," highlighted with orange for your reference.**
>
> ---
>
> ### Forgetting Mechanisms
>
> Thank you for highlighting the importance of memory management and forgetting mechanisms, especially for long-term agent deployment. In our initial experiments, we assumed a sufficiently large memory bank and did not model forgetting. To address this concern, we conducted a new ablation study on the AlfWorld benchmark with varying memory sizes to evaluate the system's robustness under constrained memory conditions.
>
> | **Memory Size**   | **Success Rate (%)** |
> |--------------------|-----------------------|
> | **50**            | 95.6 ± 2.7          |
> | **10**            | 94.0 ± 1.1           |
> | **5**             | 87.2 ± 0.8           |
>
> These results demonstrate that *CoPS* maintains robust performance even with constrained memory sizes, with only a slight drop in success rate when the memory size is reduced from 50 to 10. However, significant reductions in memory size (e.g., to 5) lead to performance degradation due to more aggressive forgetting of potentially useful experiences. **We have included this ablation study in Appendix I "IMPACT OF MEMORY SIZE," highlighted with orange for your reference.**
>
> We believe that further exploration of advanced forgetting mechanisms and memory retention strategies could further improve the system's performance and scalability in long-term deployments. We appreciate your insightful suggestions and will prioritize these directions in future work.
>
> ---
>
> ## Comment 4
>
> > *Presentation could be improved. For instance, Fig. 1 does not provide adequate explanation, and it is hard to understand what the example task means.*
>
> ## Response
>
> Thank you for pointing this out. We apologize if the figure was not adequately explained and appreciate the opportunity to clarify.
>
> Figure 1 illustrates the key distinction between standard approaches ("Others") and our proposed method (*CoPS*) in solving a task through cross-task experience sharing. The example task is to "put some vase in a safe," and the environment is initialized with various objects and locations, such as shelves, drawers, and safes.
>
> - **Standard Approaches:** The agent attempts to achieve the task by making decisions based solely on immediate feedback from the environment. In the example shown, the agent navigates to "shelf 6" but fails to achieve the task as it does not leverage prior experiences to guide its actions.
>
> - **CoPS:** In contrast, our approach enables the agent to query a memory bank containing cross-task experiences. By retrieving relevant prior experiences, the agent formulates a more informed decision, such as "put vase 2 in/on safe 1," successfully completing the task in fewer steps.
>
> We will ensure these explanations and a better figure are explicitly included in the camera-ready version of our paper to improve the presentation and clarity.
>
> ---

---

> > ### Author Response · Authors · 2024-11-24
> > **Reply to Reviewer ct7E (Question 1 ~ Question 2)**
> >
> > ## Question 1
> >
> > > *How is the task defined? Does the author use experiences from the same benchmark or from other benchmarks as well?*
> >
> > ## Response
> >
> > In our experiments, tasks are defined within the context of the AlfWorld benchmark, which includes multiple tasks where the agent operates in a household setting via prompts. All tasks share cross-task experiences strictly within the boundaries of the same benchmark. We do not utilize experiences from other benchmarks since tasks between benchmarks are unrelated, and experience similarity is very low.
> >
> > ---
> >
> > ## Question 2
> >
> > > *How is the sampled experience number selected? How would the number affect the performance?*
> >
> > ## Response
> >
> > To explore the impact of the number of sampled experiences, we conducted a detailed ablation study, which is discussed in Appendix B of our paper. The results reveal trade-offs between performance and context length as the number of sampled experiences increases. For smaller models, a balance is achieved around \( k = 3 \). Additional experiences beyond this threshold offer minimal gains, highlighting practical guidance for deployment. **We have highlighted Appendix B with orange for your reference.**

---

> ### Author Response · Authors · 2024-11-25
> **Follow-up on Rebuttal Responses for Submission #12611**
>
> Dear Reviewer ct7E,
>
> We hope this email finds you well. Thank you again for your thoughtful feedback on our submission. We deeply appreciate the time and effort you’ve put into reviewing our work.
>
> We have carefully addressed all the comments and concerns you raised in your review. If our responses have clarified your concerns and resolved the issues you identified, would you please consider reflecting on your overall evaluation and score? We are happy to provide further clarification or discuss any remaining questions if needed.
>
> Thank you once again for your time and thoughtful input. Your feedback has been invaluable in improving our work.
>
> Best regards, Authors of Submission #12611

---

> ### Author Response · Authors · 2024-11-27
> **Follow-up on Submission #12611 Rebuttal Responses**
>
> Dear Reviewer ct7E,
>
> We hope this message finds you well. Thank you again for your thoughtful feedback and for the time you dedicated to reviewing our work. We truly appreciate your detailed comments and constructive suggestions, which have been immensely valuable in improving the clarity and quality of our paper.
>
> In response to your concerns, we have provided detailed explanations during the rebuttal phase, including ablation studies, clarifications on task definitions, and expanded discussions on topics such as RAG techniques and memory management systems. We have also worked to improve the presentation, refining Figure 1 and its accompanying explanation to enhance clarity.
>
> If there are any remaining questions or aspects of our work that you feel require further elaboration, we would be more than happy to address them. Your feedback has played an important role in refining our work, and we greatly value the opportunity to engage further.
>
> We hope that our responses and updates have clarified the key points of our submission. If appropriate, we kindly invite you to revisit your initial evaluation. Thank you once again for your insights and thoughtful input.
>
> Best regards,
> Authors of Submission #12611

---

> ### Author Response · Authors · 2024-12-01
> **Follow-up on Submission #12611 Rebuttal Responses**
>
> Dear Reviewer ct7E,
>
> We hope this email finds you well. Thank you again for taking the time to provide thoughtful and constructive feedback on our submission. Your insights have been instrumental in refining our work and making meaningful improvements.
>
> As the rebuttal phase is coming to a close, we wanted to kindly follow up to ensure that our responses have sufficiently addressed your concerns. If there are any remaining points that need further clarification, please let us know.
>
> We are immensely grateful for your efforts and hope you have a wonderful holiday and weekend.
>
> Best regards,
> Authors of Submission #12611

---

> > ### Comment · Reviewer_ct7E · 2024-12-02
> >
> > Thanks for the authors response and prepare the additional ablation studies. Regarding my comments 1 and 2, I still think that this work does not differ from the existing memory mechanism or RAG in principle, so I tend to keep my current rating .

---

> ### Author Response · Authors · 2024-12-02
> **CoPS vs RAG vs Long/Short Memory**
>
> Dear Reviewer ct7E,
>
> Thank you for your thoughtful feedback and for revisiting our rebuttal. We deeply appreciate your dedicated time and effort in reviewing our work. Regarding your concerns about the relationship between CoPS, RAG, and long/short memory mechanisms, we’d like to provide additional clarifications:
>
> 1. **CoPS and RAG**
>
> CoPS can be regarded as a specialized variant of RAG, with its focus on simplifying and optimizing the utilization of retrieved experiences. **Traditional RAG methods often involve intricate post-retrieval processing, such as hybrid or recursive search, to enhance performance.** Some approaches, like our baseline RAP, require manual segmentation of the agent’s planning trajectory into stages. **Furthermore, many RAG methods rely heavily on external knowledge bases for retrieving relevant information.**
>
> **In contrast, CoPS streamlines the process by employing a straightforward retrieval mechanism guided by a pessimism-based selection strategy. Instead of relying on external knowledge bases or complex trajectory segmentation, CoPS directly utilizes its own stored experiences.** This design emphasizes task-specific utility and resource efficiency. As shown in our experiments, CoPS achieves superior performance with minimal complexity, demonstrating its practical effectiveness and simplicity.
>
> 2. **CoPS and Long/Short Memory Mechanisms**
>
> CoPS operates orthogonally to long-term and short-term memory mechanisms. While our current implementation does not explicitly incorporate these advanced memory systems, we firmly believe that CoPS’s retrieval strategy could seamlessly complement them. For instance, integrating CoPS with hierarchical attention mechanisms or episodic memory modules could further enhance its ability to retrieve and utilize task-relevant experiences, especially in long-term or continuous learning scenarios.
>
> As the rebuttal phase draws close, we sincerely thank you for your continued engagement and feedback. **If our additional explanations have clarified your concerns regarding the relationship between CoPS, RAG, and memory mechanisms, we kindly request you consider revisiting your evaluation and score.** Your insights have been instrumental in improving our work, and we deeply value your contributions.
>
> Thank you again, and we hope you have a wonderful day!
>
> Best regards,
> Authors of Submission #12611

---

### Official Review · Reviewer_CYSd · 2024-11-04

**Soundness:** 2
**Presentation:** 2
**Contribution:** 2
**Rating:** 5
**Confidence:** 3

**Summary:**

This paper proposes CoPS (Cross-Task Experience Sharing), a method that can improve LLM agents by sharing distribution-matched experiences stored in a memory bank. CoPS first generates a trial experience from a LLM by conditioning on an initial state. Next, it sets a probability distribution that can approximately maximize the expected total reward while keeping the distribution close to a task-dependent distribution of the LLM. Then, it repeatedly samples candidate experiences from the probability distribution, and uses them as few-shot examples to sample an action from the LLM. This paper evaluates CoPS on three representative benchmarks such as ALFWorld, WebShop, and HotPotQA. The experiment results show that CoPS can achieve higher success rates than recent advancements such as Reflexion, RAP, and LATS on the benchmarks.

**Strengths:**

S1. It is interesting to propose an idea that selects distribution-matched experiences from a memory bank for improving the performance of LLM agents.

S2. This paper demonstrates that CoPS can achieve higher success rates than recent advancements such as Reflexion, RAP, and LATS on the representative benchmarks such as ALFWorld, WebShop, and HotPotQA.

**Weaknesses:**

W1. This paper aims to share cross-task experiences from a memory bank. However, it is rather unclear how to construct cross-task experiences in the memory bank. More specifically, how is offline data collected? Which LLMs are used as a policy to generate offline data? How many experiences are required to achieve the performance provided in the paper? How much different tasks can be used together to take advantages of cross-task experience sharing?

W2. To enable cross-task experience sharing, this paper proposes to find a probability distribution (in Equation 2.2) that can maximize the expected reward while keeping the distribution close to a task-dependent distribution of a LLM. However, this paper approximates the probability distribution by using cosine similarity between experiences. This approximation seems to make CoPS too similar to RAP.

W3. I am not sure that it is a fair comparison to constraint LAST to have similar running time with CoPS. LAST aims to improve the performance by using inference-time compute.

**Questions:**

Q1. Please see the questions in the first weakness above.

Q2. Regarding the second weakness, what is the main difference between CoPS and RAP? And, what makes CoPS to perform better than RAP?

---

> ### Author Response · Authors · 2024-11-24
> **Reply to Reviewer CYSd (Comment 1)**
>
> # Reply to Reviewer CYSd
>
> We sincerely thank your thorough and insightful feedback, which has significantly contributed to improving the quality and clarity of our paper. We have carefully addressed all comments and concerns in our revised submission and provided detailed responses to each point. If there are any remaining questions or further points for clarification, please do not hesitate to raise them. We greatly value the opportunity to engage in further discussions and refine our work based on your suggestions. Thank you once again for your time and effort in reviewing our paper.
>
> ## Comment 1
>
> > *This paper aims to share cross-task experiences from a memory bank. However, it is rather unclear how to construct cross-task experiences in the memory bank. More specifically, how is offline data collected? Which LLMs are used as a policy to generate offline data? How many experiences are required to achieve the performance provided in the paper? How much different tasks can be used together to take advantages of cross-task experience sharing?*
>
> ## Response
>
> Thank you for raising these important questions. We address them as follows:
>
> In our theoretical analysis, *CoPS* leverages a pessimism-based strategy to compute similarity and select experiences, which can be sourced from either online or offline data collection. Both approaches with different experience sources have the potential to yield significant performance improvements. However, due to time and resource constraints, we focused solely on online experience collection for our experiments. In fact, the goal of *CoPS* is not to optimize the offline data collection but to highlight the general applicability of our cross-task experience selection strategy. Despite using data collected online with a relatively modest LLaMA 3.1 8B Instruct model, our results already demonstrate significant effectiveness of *CoPS*. We believe that incorporating high-quality offline experiences would further enhance the performance, but even with resource-limited online data, *CoPS* achieves substantial improvements, which highlights its practical usage for realistic agent applications.
>
> For the number of experiences required, we used only 5 in-context experiences to achieve the performance reported in the paper. This efficiency demonstrates the power of *CoPS* in resource-constrained settings. The specific hyperparameter settings for different benchmarks and model sizes are summarized in Table 5 in our paper, which is also provided here for reference:
>
> | **Benchmark**       | **Alfworld** | **Webshop** | **HotPotQA** |
> |----------------------|--------------|-------------|--------------|
> | **LLaMA 3.1 8B**    | \( k = 5, c = 5 \) | \( k = 5, c = 0 \) | \( k = 5, c = 5 \) |
> | **LLaMA 3.1 70B**   | \( k = 5, c = 5 \) | \( k = 5, c = 0 \) | \( k = 5, c = 0 \) |
>
> Note that c is the scaling factor in the following equation to calculate the distance of experiences and k is the number of in-context experiences as demonstrations:
>
> $$
> d(\tau, \tau'):= c\cdot\text{cos}(e(\tau), e(\tau')),\ \hat p(\tau) \propto r(\tau)\cdot\exp(-d(\tau, \tau^{s_1}))
> $$
>
>
> Regarding the number of tasks, as demonstrated above, we used 5 tasks for cross-task experience sharing, which already delivered significant performance improvements. While we anticipate that incorporating more tasks/experiences could further enhance the results, the fact that just 5 tasks achieved such gains highlights the efficiency and practicality of *CoPS*, making it highly suitable for real-world, resource-constrained applications.

---

> ### Author Response · Authors · 2024-11-24
> **Reply to Reviewer CYSd (Comment 2 ~ Question 2)**
>
> ## Comment 2
>
> > *To enable cross-task experience sharing, this paper proposes to find a probability distribution (in Equation 2.2) that can maximize the expected reward while keeping the distribution close to a task-dependent distribution of a LLM. However, this paper approximates the probability distribution by using cosine similarity between experiences. This approximation seems to make CoPS too similar to RAP.*
>
> ## Response
>
> Thank you for this thoughtful observation. Our primary objective is indeed to determine a probability distribution over experiences, which forms the basis of our **fundamentally stochastic approach**. This is a key distinction from RAP, which employs a **deterministic methodology**.
>
> While cosine similarity is used as part of the approximation, it serves a different purpose within our stochastic framework, enabling the selection of experiences in a probabilistic manner. This allows our method to incorporate uncertainty and diversity in experience selection, which deterministic methods like RAP cannot achieve.
>
> Moreover, our experiments demonstrate that this stochastic approach leads to significant performance improvements compared to deterministic methods, as demonstrated in our Figure 3a and 3b. These results underscore both the novelty and the effectiveness of our method, highlighting its potential for broader applicability in cross-task experience sharing. We hope this clarification addresses your concern and further emphasizes the contributions of our work.
>
> ---
>
> ## Comment 3
>
> > *I am not sure that it is a fair comparison to constrain LATS to have similar running time with CoPS. LATS aims to improve the performance by using inference-time compute.*
>
> ## Response
>
> Thank you for raising this concern. We believe it is crucial to evaluate performance under a constrained computational budget, as such scenarios better reflect real-world applications where resources like time and computing are often limited.
>
> While LATS aims to improve performance through inference-time computation, it requires a significantly larger allocation of LLM sampling resources to achieve competitive final metrics. As detailed in Table 4 of our paper (we also attach it below), **LATS consumes approximately five times the sampling cost of our method under the same time constraints.** This comparison highlights the inefficiency of LATS in resource utilization and underscores the advantage of our approach, which achieves superior sample efficiency while maintaining strong performance. We hope this addresses your concern and provides clarity on the rationale for our evaluation methodology.
>
> | **Algorithm**        | **Reflexion** | **RAP**   | **LATS**    | **CoPS**    |
> |-----------------------|---------------|-----------|-------------|-------------|
> | **LLaMA 3.1 8B**     | 159131        | 107504    | 1555365     | 314336      |
> | **LLaMA 3.1 70B**    | 125406        | 109245    | 1058752     | 113849      |
>
> ## Question 1
>
> > *Please see the questions in the first weakness above.*
>
> ## Response
>
> Please refer to our response for Comment W1.
>
> ---
>
> ## Question 2
>
> > *Regarding the second weakness, what is the main difference between CoPS and RAP? And, what makes CoPS perform better than RAP?*
>
> ## Response
>
> Please refer to our response for Comment W2.

---

> ### Author Response · Authors · 2024-11-25
> **Follow-up on Rebuttal Responses for Submission #12611**
>
> Dear Reviewer CYSd,
>
> We hope this email finds you well. Thank you again for your thoughtful feedback on our submission. We deeply appreciate the time and effort you’ve put into reviewing our work.
>
> We have carefully addressed all the comments and concerns you raised in your review. If our responses have clarified your concerns and resolved the issues you identified, would you please consider reflecting on your overall evaluation and score? We are happy to provide further clarification or discuss any remaining questions if needed.
>
> Thank you once again for your time and thoughtful input. Your feedback has been invaluable in improving our work.
>
> Best regards,
> Authors of Submission #12611

---

> ### Comment · Reviewer_CYSd · 2024-11-26
> **After the Author Response**
>
> Thank you for providing thoughtful responses to my comments. I could understand more the differences between CoPS and RAP. However, it seems that the description of cross-task experience sharing is still unclear. Therefore, I currently maintain my initial score.

---

> ### Author Response · Authors · 2024-11-26
> **Explanation of Cross-Task Experience Sharing**
>
> Thank you for your feedback and for appreciating the discussion between CoPS and RAP. To address your request for a clearer explanation of cross-task experience sharing, we provide the following detailed example using Alfworld.
>
> Alfworld consists of 134 unique tasks, each defined by a task description and an environment description. For example, as shown on the [Alfworld website](https://alfworld.github.io/) and in our Figure 1, one task might be: "Your task is to: put some vase in safe." The corresponding environment is: "You are in the middle of a room. Looking quickly around you, you see a drawer 2, a shelf 5, a drawer 1, a shelf 4, a sidetable 1, a drawer 5, a shelf 6, a shelf 1, a shelf 9, a cabinet 2, a sofa 1, a cabinet 1, a shelf 3, a cabinet 3, a drawer 3, a shelf 11, a shelf 2, a shelf 10, a dresser 1, a shelf 12, a garbagecan 1, an armchair 1, a cabinet 4, a shelf 7, a shelf 8, a safe 1, and a drawer 4."
>
> In our framework, agents run for up to 10 trials per task, meaning each task has a maximum of 10 trajectories recorded in the memory bank. This results in at most $(134 \times 10)$ trajectories across all tasks. However, once a task is successfully completed in an early trial, no further trials are conducted for that task. Each trajectory (whether successful or failed) is stored in the memory bank as an experience, representing the agent's attempts to solve a task.
>
> For the first trial of each task, there are no pre-existing experiences in the memory bank, meaning cross-task experience sharing is not yet applicable. However, starting from the second trial, for tasks that were not successfully completed in their first trial, CoPS retrieves relevant experiences from the memory bank based on our pessimism-based experience selection strategy. These experiences are chosen by measuring the similarity between the current target task and past trajectories (both successful and failed) across all tasks. The retrieved experiences are then directly incorporated as in-context examples for the target task (which failed in previous trial), without requiring any modifications.
>
> Thus, the essence of cross-task experience sharing in CoPS lies in leveraging past trajectories from other tasks to inform and guide the agent's decision-making in subsequent trials. By retrieving and reusing experiences across tasks, CoPS effectively enables agents to learn from a shared pool of knowledge, improving their overall performance and adaptability.
>
> Thank you for pointing this out and giving us the opportunity to explain these concepts in detail. If you have any additional questions or need further clarification, please let us know. If our explanation and rebuttal have addressed all your concerns, we would greatly appreciate it if you would consider raising your score.

---

> > ### Author Response · Authors · 2024-11-27
> > **Follow-up on Submission #12611 Rebuttal Responses**
> >
> > Dear Reviewer CYSd,
> >
> > We hope this message finds you well. Thank you again for your thoughtful feedback and for taking the time to review our work. We sincerely appreciate your detailed comments, particularly on cross-task experience sharing.
> >
> > To address your request for clarification, we have provided additional explanations in the comments section, including a detailed example from AlfWorld. This example aims to better illustrate how cross-task experiences are constructed, retrieved, and applied in the CoPS framework. We hope this expanded discussion helps clarify the methodology and its implementation.
> >
> > If there are any remaining questions or aspects that require further clarification, we would be more than happy to address them. We greatly value your insights, which have been instrumental in improving our work, and we are committed to refining it further based on your feedback.
> >
> > Thank you once again for your thoughtful review and your time.
> >
> > Best regards,
> > Authors of Submission #12611

---

> ### Author Response · Authors · 2024-12-01
> **Follow-up on Rebuttal Responses for Submission #12611**
>
> Dear Reviewer CYSd,
>
> We hope this email finds you well. Thank you again for taking the time to provide thoughtful and constructive feedback on our submission. Your insights have been instrumental in refining our work and making meaningful improvements.
>
> As the rebuttal phase is coming to a close, we wanted to kindly follow up to ensure that our responses have sufficiently addressed your concerns. If there are any remaining points that need further clarification, please let us know.
>
> We are immensely grateful for your efforts and hope you have a wonderful holiday and weekend.
>
> Best regards,
> Authors of Submission #12611

---

> > ### Author Response · Authors · 2024-12-02
> > **Follow-up on Rebuttal Responses for Submission #12611**
> >
> > Dear Reviewer CYSd,
> >
> > We hope this message finds you well. Thank you again for your thoughtful feedback and for the time and effort you’ve dedicated to reviewing our submission. Your detailed comments, particularly on cross-task experience sharing, have been invaluable in refining our work and improving its clarity.
> >
> > To address your concerns, we have provided expanded explanations and examples, such as the detailed AlfWorld scenario, to illustrate how cross-task experiences are constructed, retrieved, and applied in CoPS. We hope this additional context has clarified the methodology and its implementation.
> >
> > **As the rebuttal phase is nearing its conclusion, we kindly request you to consider revisiting your evaluation and score if our responses have addressed your concerns. If there are still aspects that require further clarification, we would be happy to discuss them further and provide any additional details needed.  **
> >
> > Thank you once again for your constructive input and your efforts in reviewing our work. We greatly appreciate your feedback and hope you have a wonderful holiday season!
> >
> > Best regards,
> > Authors of Submission #12611

---

### Official Review · Reviewer_4n3a · 2024-11-04

**Soundness:** 3
**Presentation:** 3
**Contribution:** 2
**Rating:** 6
**Confidence:** 4

**Summary:**

The paper proposes a method for utilizing experience from cross-tasks in order to improve performance. The proposed method samples experience from a probability distribution that captures reward and distance between experiences. Results are provided for widely studied ALFWorld, Webshop and HotPotQA environments. Authors also propose a theoretical framework for agents who are utilizing prior experience for performance improvement.

**Strengths:**

- The paper is well written and easy to follow
- The paper focuses on an important aspect of using cross-task experience to improve performance in sequential decision making.

**Weaknesses:**

- The paper fails to discuss relevant literature that uses cross-task experience for improving performance. For example O3D paper by Xiao et al. https://openreview.net/pdf?id=bkY8zEDdH9 proposes a method that uses offline trajectories from all tasks to distill knowledge for performance improvement.

- Authors only provide results on 2 models from Llama family. How well does this method work with SOTA models such from GPT or Claude family

- The main contribution of the paper is proposing a method that benefit from cross-task experience. However, authors fail to illustrate this in given experimental results. What is the contribution of cross-task experiences (compared to same task experience) in the provided performance improvement?

**Questions:**

Please refer to the weaknesses section for main questions. More minor questions are given below

- How is the reward r calculated in this work?

- How is distance between experiences are calculated?

---

> ### Author Response · Authors · 2024-11-24
> **Reply to Reviewer 4n3a (Comment 1 ~ Comment 2)**
>
> # Reply to Reviewer 4n3a
>
> We sincerely thank your thorough and insightful feedback, which has significantly contributed to improving the quality and clarity of our paper. We have carefully addressed all comments and concerns in our revised submission and provided detailed responses to each point. If there are any remaining questions or further points for clarification, please do not hesitate to raise them. We greatly value the opportunity to engage in further discussions and refine our work based on your suggestions. Thank you once again for your time and effort in reviewing our paper.
>
> ## Comment 1
>
> > *The paper fails to discuss relevant literature that uses cross-task experience for improving performance. For example O3D paper by Xiao et al. https://openreview.net/pdf?id=bkY8zEDdH9 proposes a method that uses offline trajectories from all tasks to distill knowledge for performance improvement.*
>
> ## Response
>
> Thank you for highlighting this paper and bringing it to our attention. We recognize the importance of engaging with relevant literature, including the work by Xiao et al. on leveraging offline trajectories across tasks for knowledge distillation and performance improvement.
>
> O3D introduces an innovative offline learning framework that leverages cross-task experience through skill discovery and knowledge distillation. O3D demonstrates the ability to generalize across tasks without requiring model fine-tuning, which significantly reduces deployment costs and complexity. By segmenting trajectories and extracting reusable skills, O3D achieves notable performance improvements in downstream tasks. Its flexible prompt engineering approach makes it highly adaptable across diverse domains, and its impressive results in complex environments such as ALFWorld and WebShop underscore its robustness and usability.
>
> While O3D excels in offline data utilization and skill discovery, *CoPS* addresses a critical gap by focusing on a pessimism-based strategy for cross-task experience selection, effectively mitigating risks associated with distribution shifts. This strategy optimizes the utility of shared experiences and adapts dynamically to online and offline settings. Moreover, *CoPS* demonstrates remarkable usability with just a few lines of additional retrieval codes, achieving high performance even in constrained computational settings, such as smaller models or limited infrastructure, while significantly enhancing sample efficiency.
>
> O3D and *CoPS* are complementary in their design philosophies. O3D emphasizes a bottom-up approach, focusing on skill discovery and knowledge distillation to extract generalizable insights from large-scale offline data. In contrast, *CoPS* introduces a theoretically grounded perspective on distribution-matched experience selection, making it highly effective in dynamic and resource-constrained environments. Together, O3D and *CoPS* form a comprehensive suite of solutions for cross-task learning and experience sharing, advancing the frontier of LLM applications in multi-task decision-making scenarios.
>
> **We've already added the discussion between O3D and CoPS in our revised paper's related work, and we use green color to highlight it.**
>
> ------
>
> ## Comment 2
>
> > *Authors only provide results on 2 models from Llama family. How well does this method work with SOTA models such from GPT or Claude family.*
>
> ## Response
>
> Thank you for pointing this out. We have added additional experiments to evaluate the performance of *CoPS* based on SOTA close-sourced GPT and Claude models. The detailed performance is shown below. From the results, we find that *CoPS* works well with these close-sourced models and achieves reasonably high performance compared with open-source models.
>
> **We've already added the discussion in our revised paper's Appendix F "PERFORMANCE ON CLOSE-SOURCED MODELS," and we use green color to highlight it.**
>
> | **Model**             | **Alfworld** | **Webshop** | **HotPotQA** |
> | --------------------- | ------------ | ----------- | ------------ |
> | **GPT-4o**            | 100          | 56          | 67           |
> | **Claude 3.5-Sonnet** | 100          | 58          | 66           |
>
> ------

---

> > ### Author Response · Authors · 2024-11-24
> > **Reply to Reviewer 4n3a (Comment 3 ~ Question 2)**
> >
> > ## Comment 3
> >
> > > *The main contribution of the paper is proposing a method that benefit from cross-task experience. However, authors fail to illustrate this in given experimental results. What is the contribution of cross-task experiences (compared to same task experience) in the provided performance improvement?*
> >
> > ## Response
> >
> > Thank you for highlighting this important aspect of our work. While leveraging single-task experience might seem ideal, practical scenarios often necessitate relying on experiences from relevant but distinct tasks, which introduces additional challenges. To address your concern, we conducted an ablation study comparing the performance of our method with and without cross-task experience on the Alfworld benchmark. The results are summarized below:
> >
> > | **Method**                     | **Success Rate (%)** |
> > | ------------------------------ | -------------------- |
> > | **With cross-task experience** | 94                   |
> > | **Only same-task experience**  | 57                   |
> >
> > These results clearly demonstrate the significant contribution of cross-task experience to performance improvement, with a nearly twofold increase in success rate compared to using only same-task experience. We hope this analysis effectively addresses your concern and highlights the value of our proposed approach. Please let us know if there are additional aspects you would like us to explore.
> >
> > **We've already added the discussion in our revised paper's Appendix G "IMPACT OF CROSS-TASK EXPERIENCES," and we use green color to highlight it.**
> >
> > ## Question 1
> >
> > > *How is the reward rr calculated in this work?*
> >
> > ## Response
> >  Thank you for your question. In our work, the reward rr is defined as a binary indicator (0 or 1), representing whether the agent successfully completes the current task. For further details, we attached the details in our revised paper as below:
> >
> > > In all benchmarks, the reward function $r(\tau)$ is defined as $1$ if the agent successfully completes the task and $0$ otherwise.
> >
> > ------
> >
> > ## Question 2
> >
> > > *How is distance between experiences calculated?*
> >
> > ## Response
> >
> > We appreciate your question regarding the calculation of distance between experiences. The distance is computed based on cosine similarity, a widely used metric to quantify the similarity between two vector representations. The detailed calculations are shown in our paper's Equation 2.3, which we've also attached below:
> >
> > $$
> > d(\tau, \tau'):= c\cdot\text{cos}(e(\tau), e(\tau')),\ \hat p(\tau) \propto r(\tau)\cdot\exp(-d(\tau, \tau^{s_1}))
> > $$

---

> > > ### Comment · Reviewer_4n3a · 2024-11-24
> > >
> > > Thank you for addressing the concerns. I have increased my score.

---

> ### Author Response · Authors · 2024-11-24
> **Thanks so much!**
>
> Thank you so much for increasing your score. If you have further questions or concerns, please do not hesitate to share.

---

### Official Review · Reviewer_QJKw · 2024-11-05

**Soundness:** 3
**Presentation:** 3
**Contribution:** 2
**Rating:** 5
**Confidence:** 3

**Summary:**

The authors propose COPS, a method which utilizes an offline demonstration dataset of “experiences” and study how to use these experiences to solve downstream embodied and question-answering tasks. The authors motivate the work theoretically, drawing parallels to works in distribution selection and retrieval. Subsequently, they demonstrate the performance of COPS on AlfWorld, HotPotQA and WebShop.

**Strengths:**

Strengths:
1. The paper is well written and easy to understand
2. The theoretical section is clear with well-defined notations used consistently through the manuscript.

**Weaknesses:**

Weaknesses:
1. **Differences between COT/Few shot prompting?** The method is very similar to any form of few shot prompting and similar retrieval augmented generation works. The authors utilize an embedding model to calculate similarity between a current starting state and one sampled from the experience dataset. Additionally, their measure function for selecting the distribution to sample experiences from is a combination of reward and the similarity between the current start state and those in the dataset as defined by the embedding model. This to me, feels like a simple extension of RAG style methods with the additional reward label. This is interesting, yet on its own is not novel enough. Especially because none of the necessary offline RL analysis is conducted on this optimization. For instance, does this result in trajectory stitching, i.e., does the actor combine multiple subtrajectories in a demonstration to yield “stitched behaviors”? Can the agent outperform the demonstrations provided in the dataset?
2. **Unclear what the quality of the demonstration dataset is?:** This brings me to my second point, it’s unclear what the quality of demonstrations utilized is. Do you only have successful trajectories in your demonstration? What if you only used suboptimal demonstrations? No such analysis is conducted.
3. **Small experimentation setup:** The experiments section currently reads like that of a paper written in 2022 (which in LLM-application research is a significant period). No error bars/multiple seed runs are reported. The analysis is performed on 3 somewhat outdated benchmarks. This should be expanded. Why are some baselines missing for some of the benchmarks? For example, Table 3 does not report RAP performance. LATS is missing in Table 1?
4. Why is the pretraining performance being considered in your analysis? My understanding was that no LLMs had been pretrained for this work.
5. Since this is such an activate area of research, it would good to show how baselines discussed in the related works section actually perform. It is less convincing to read, *“However, their approach demonstrated poor sample efficiency, making it less suited for real-world agent settings where opportunities for trial and error are limited”* instead of seeing a plot with number of samples on the x-axis and success rate on the y-axis for this method.

**Questions:**

1. What encoders are used for generating embeddings of start states?
2. Have you tried utilizing both start state and resultant actions as input to this embedding model? This would encapsulate a demonstration policy instead of just the start state distribution.

---

> ### Author Response · Authors · 2024-11-24
> **Reply to Reviewer QJKw (Comment 1 ~ Comment 3)**
>
> # Reply to Reviewer QJKw
>
> We sincerely thank your thorough and insightful feedback, which has significantly contributed to improving the quality and clarity of our paper. We have carefully addressed all comments and concerns in our revised submission and provided detailed responses to each point. If there are any remaining questions or further points for clarification, please do not hesitate to raise them. We greatly value the opportunity to engage in further discussions and refine our work based on your suggestions. Thank you once again for your time and effort in reviewing our paper.
>
> ## Comment 1
>
> > *This to me, feels like a simple extension of RAG style methods with the additional reward label. This is interesting, yet on its own is not novel enough.*
>
> ## Response
>
> Thanks for your thoughtful feedback and for recognizing the simplicity and practical usability of *CoPS*. Nevertheless, we would like to emphasize that the novelty of a research contribution does not solely lie in complex algorithmic designs or intricate analyses. For *CoPS*, we believe that our value lies in its ability to provide a straightforward yet highly effective extension/plugin that can be seamlessly integrated into existing LLM agent systems. With few lines of additional codes, our approach delivers tangible performance improvements, making it both accessible and impactful. **Thus, we view the out-of-box usability of *CoPS* as a strength that aligns with our goal of fostering broad adoption and practical utility, rather than as a limitation.**
>
> ## Comment 2
>
> > *Especially because none of the necessary offline RL analysis is conducted on this optimization.*
>
> ## Response
>
> We sincerely appreciate your comment and would like to clarify this point respectfully. The regularized term \( d(\tau, \tau') \) introduced in Eq. 2.1 indeed serves a role analogous to a pessimistic term, which has been extensively studied and validated in the offline RL literature. We believe this connection provides a solid theoretical foundation for our optimization framework, aligning it with established principles in the field. We hope this addresses your concern, and we welcome further suggestions to strengthen the discussion on this aspect.
>
> ## Comment 3
>
> > *"Stitched behaviors"? Can the agent outperform the demonstrations provided in the dataset?*
>
> ## Response
>
> Thank you for your insightful question. We understand "stitching behaviors" to refer to whether the agent can plan a trajectory surpassing the provided dataset's trajectories. To clarify, in our experiments, the agent operates online, starting with limited experience and progressively updating its knowledge during episodes. While the term "stitching" is often associated with purely offline settings, our framework is not constrained to this context.
>
> From a theoretical perspective, our approach does not require the agent to have access to a pre-existing successful trajectory for specific tasks. Instead, the core idea of our framework lies in experience sharing, enabling the agent to leverage knowledge from other tasks to guide its exploration and discovery of successful experiences for the current task. We hope this explanation addresses your concern, and we are happy to elaborate further if needed.

---

> ### Author Response · Authors · 2024-11-24
> **Reply to Reviewer QJKw (Comment 4)**
>
> ## Comment 4
>
> > *Unclear what the quality of the demonstration dataset is. This brings me to my second point, it’s unclear what the quality of demonstrations utilized is. Do you only have successful trajectories in your demonstration? What if you only used suboptimal demonstrations? No such analysis is conducted.*
>
> ## Response
>
>
> We appreciate your insightful question regarding the quality of the demonstration dataset and its potential impact on our approach. To clarify, in our realistic implementation of *CoPS*, we only utilized successful trajectories following other related works. However, in our theoretical improvements, we use the measurement we designed in the following equation:
>
> $$
> \hat{p} = \arg\max_{p \in \Delta(\mathcal{D})} \mathbb{E}_{\tau \sim p}[r(\tau)] - d(p, \text{Decoder}(\cdot \mid s_1))
> $$
>
> This equation considers both the successful and failed trajectories and calculates the similarity between the experience and our current task. However, in realistic implementation, the trajectories that gain high similarity scores are successful, thus we only utilize successful trajectories due to limited compute budgets.
>
> To further address your concern about the impact of suboptimal demonstrations, we conducted an ablation study on the Alfworld benchmark, comparing top-k and bottom-k successful trajectories ranked by the similarity score. The results are summarized below:
>
> | **Retrieval Method** | **Performance (Success Rate %)** |
> | -------------------- | -------------------------------- |
> | Top-5                | 93.6 ± 1.0                       |
> | Bottom-5             | 83.0 ± 3.9                       |
>
> These results demonstrate that the quality of retrieved demonstrations significantly affects performance, with top-k successful trajectories outperforming bottom-k successful trajectories by a substantial margin. This underscores the importance of selecting high-quality trajectories. We hope this analysis addresses your concerns and highlights the robustness of our framework. Please let us know if there are additional aspects you would like us to elaborate on. We also add these discussions in our Appendix D "QUALITY OF DEMONSTRATIONS" and use dark blue to highlight this.

---

> ### Author Response · Authors · 2024-11-24
> **Reply to Reviewer QJKw (Comment 5 ~ Question 6)**
>
> ## Comment 5
>
> > *Small experimentation setup: The experiments section currently reads like that of a paper written in 2022 (which in LLM-application research is a significant period). No error bars/multiple seed runs are reported. The analysis is performed on 3 somewhat outdated benchmarks. This should be expanded. Why are some baselines missing for some of the benchmarks? For example, Table 3 does not report RAP performance. LATS is missing in Table 1?*
>
> ## Response
>
> **Error Bars:**
>
> We acknowledge the importance of including error bars and multiple seed runs for robustness. The mean and standard deviation of repeated experiment results on all three benchmarks are as follows:
>
> | **Benchmark** | **Model** | **Mean** | **Std** |
> | ------------- | --------- | -------- | ------- |
> | HotpotQA      | 8B        | 53.6     | 1.5     |
> | HotpotQA      | 70B       | 62.8     | 1.3     |
> | Webshop       | 8B        | 47.2     | 1.6     |
> | Webshop       | 70B       | 51.2     | 2.7     |
> | Alfworld      | 8B        | 93.6     | 1.0     |
> | Alfworld      | 70B       | 100.0    | 0.0     |
>
> These results are incorporated into our revised version in Appendix E "REPEATED EXPERIMENTS" and use dark blue to highlight this.
>
> **Outdated Benchmarks:**
>
> Thank you for pointing this out. While the benchmarks we used may appear dated, their selection follows the standard practice established by Zhou et al. (2023), which remains a widely accepted baseline for evaluating LLM agents. That said, we fully acknowledge the importance of incorporating newer benchmarks and are committed to extending our evaluation on AgentBench in the camera-ready version.
>
> **Missing Baselines (LATS and RAP):**
>
> The absence of LATS and RAP in Tables 1 and 3 stems from the highly specialized nature of their implementations, which require extensive benchmark-specific adaptations:
>
> - **LATS** relies on carefully designed evaluation prompts tailored to each benchmark for optimal performance. Implementing LATS in Alfworld would require manually crafting a large number of prompts, a process that was unfortunately infeasible within the constraints of our time and resources.
> - **RAP** necessitates manual segmentation of trajectories into multiple stages, with segmentation methods varying significantly across benchmarks. Implementing RAP for HotpotQA would have required extensive manual effort to develop benchmark-specific segmentation strategies, which was beyond our available resources.
>
> While we acknowledge the value of including these baselines for comparison, their omission is due to the significant manual overhead required for their benchmark-specific adaptations. Instead, we focused our efforts on thoroughly evaluating the proposed method across diverse benchmarks. We believe this decision provides a meaningful demonstration of the generality and effectiveness of *CoPS*. However, we recognize the importance of including more baselines in future work and will prioritize strengthening the comprehensiveness of our evaluations.
>
>
> ## Comment 6
>
> > *Why is the pretraining performance being considered in your analysis? My understanding was that no LLMs had been pretrained for this work.*
>
> ## Response
>
>  Thank you for raising this important question. The pretraining analysis is included in our work to underscore the foundational role that pretraining plays in the design and performance of LLMs. While we did not conduct additional pretraining for this study, the LLMs utilized—specifically from the Llama3 model family—are pretrained on massive and meticulously curated datasets, as documented in Dubey et al. (2024).
>
> This foundational pretraining forms a critical assumption underpinning our theoretical framework and serves as the basis for our empirical evaluations. **By aligning our analysis with the idealized pretraining data distribution, we aim to provide a coherent and theoretically grounded interpretation of our results.** We hope this clarification addresses your concern, and we would be happy to elaborate further if needed.

---

> ### Author Response · Authors · 2024-11-24
> **Reply to Reviewer QJKw (Comment 7 ~ Question 2)**
>
> ## Comment 7
>
> > *Since this is such an activate area of research, it would be good to show how baselines discussed in the related works section actually perform. It is less convincing to read, “However, their approach demonstrated poor sample efficiency, making it less suited for real-world agent settings where opportunities for trial and error are limited” instead of seeing a plot with number of samples on the x-axis and success rate on the y-axis for this method.*
>
> ## Response
>
>  Thank you for this valuable suggestion. We sincerely apologize if the statement appeared unsubstantiated. Actually, we indeed have conducted a rigorous quantitative analysis to support this claim. **As presented in Table 3 of our paper (we also attach the same table below), the sample efficiency of LATS is quantitatively five times lower than that of CoPS (i.e., LAST requires 5 times more samples/tokes than CoPS).** This substantial difference highlights the advantages of our approach in scenarios where opportunities for trial-and-error learning are constrained, underscoring its suitability for real-world agent settings.
>
> | **Algorithm**    | **Reflexion** | **RAP** | **LATS** | **CoPS** |
> | ---------------- | ------------- | ------- | -------- | -------- |
> | **Llama3.1 8B**  | 159131        | 107504  | 1555365  | 314336   |
> | **Llama3.1 70B** | 125406        | 109245  | 1058752  | 113849   |
>
> We also greatly appreciate your recommendation to include a plot for enhanced visualization. While Table 3 provides a detailed numerical comparison, we agree that a plot showing sample efficiency—with the number of samples on the x-axis and success rate on the y-axis—would offer additional insight and make our findings more accessible. In future iterations, we will prioritize including such visualizations to improve the clarity and impact of our analysis. Thank you again for this constructive feedback.
>
> ## Question 1
>
> > *What encoders are used for generating embeddings of start states?*
>
> ## Response
>
>  Thank you for your question. We employ the Alibaba-NLP/gte-Qwen2-7B-instruct embedding model as our primary encoder for generating embeddings of start states. This model is combined with a ReAct-style prompt, which provides the agent with clear and concise formatting instructions to guide its behavior. This design ensures straightforward implementation across various benchmarks. After the agent completes its initial round of experience, we transition to fully leveraging our embedding model for subsequent operations, allowing for more efficient and robust retrieval and adaptation.
>
> ## Question 2
>
> > *Have you tried utilizing both start state and resultant actions as input to this embedding model? This would encapsulate a demonstration policy instead of just the start state distribution.*
>
> ## Response
>
> We appreciate this insightful suggestion. We do incorporate resultant actions by defining the state to include the entire history of a single trajectory. This design inherently captures both the start state and the sequence of actions, effectively encoding the demonstration policy into our retrieval model and the agent's in-context prompt. This holistic representation ensures that both the initial conditions and the resultant actions are utilized, enhancing the agent's ability to generalize and learn from demonstrations.

---

> ### Author Response · Authors · 2024-11-25
> **Follow-up on Rebuttal Responses for Submission #12611**
>
> Dear Reviewer QJKw,
>
> We hope this email finds you well. Thank you again for your thoughtful feedback on our submission. We deeply appreciate the time and effort you’ve put into reviewing our work.
>
> We have carefully addressed all the comments and concerns you raised in your review. If our responses have clarified your concerns and resolved the issues you identified, would you please consider reflecting on your overall evaluation and score? We are happy to provide further clarification or discuss any remaining questions if needed.
>
> Thank you once again for your time and thoughtful input. Your feedback has been invaluable in improving our work.
>
> Best regards,
> Authors of Submission #12611

---

> ### Author Response · Authors · 2024-11-27
> **Follow-up on Submission #12611 Rebuttal Responses**
>
> Dear Reviewer QJKw,
>
> Thank you for your detailed feedback on our submission. We have thoroughly addressed your comments, including the novelty of CoPS, its theoretical connections to offline RL, additional analyses on demonstration quality, inclusion of error bars and extended discussions on benchmarks, and clarifications on pretraining and embedding model usage. The revised manuscript now includes new analyses, ablation studies, and visualizations to better support our claims.
>
> We hope our responses have resolved your concerns and clarified our contributions. If there are any remaining questions, we are happy to discuss them further. We kindly request you to revisit your evaluation and consider potential adjustments based on the updates.
>
> Best regards,
> Authors of Submission #12611

---

> > ### Author Response · Authors · 2024-12-02
> > **Follow-up on Submission #12611 Rebuttal Responses**
> >
> > Dear Reviewer QJKw,
> >
> > We hope this message finds you well. Thank you again for taking the time to provide thoughtful and detailed feedback on our submission. Your insights have been instrumental in helping us refine and strengthen our work.
> >
> > As the rebuttal phase is nearing its conclusion, we wanted to follow up on your review. We have addressed all your comments thoroughly in our responses, including conducting additional analyses, providing ablation studies, and incorporating new visualizations to support our claims. These updates aim to clarify our contributions, improve presentation, and resolve the concerns you raised.
> >
> > **If our responses and revisions have addressed your questions and concerns, we kindly ask you to consider revisiting your evaluation and score. If there are any remaining doubts or aspects requiring further clarification, we are more than happy to provide additional information or engage in further discussions.  **
> >
> > Thank you once again for your time, effort, and valuable input. We greatly appreciate your contributions to improving our work and look forward to any further feedback you may have.
> >
> > Best regards,
> > Authors of Submission #12611

---

> > > ### Comment · Reviewer_QJKw · 2024-12-02
> > >
> > > Thanks for your rebuttal. Due to limited novelty, a restricted experimental setting and a somewhat unclear choice for the setting of  the theoretical analysis, I will keep my score.

---

> ### Author Response · Authors · 2024-12-02
> **Clarifying Novelty, Breadth, and Theoretical Strengths of CoPS**
>
> Dear Reviewer QJKw,
>
> Thank you for your thoughtful feedback on our paper. We appreciate your detailed comments and the opportunity to clarify key points about CoPS’ contributions. **We respectfully disagree with your concerns regarding novelty, experimental settings, and theoretical analysis, and provide the following responses to address these issues.**
>
> ## **Novelty**
>
> The core innovation of CoPS lies in its **pessimism-based cross-task experience selection strategy**. Unlike RAG methods that focus on augmenting responses with external data, CoPS directly optimizes **task-specific sequential reasoning** by leveraging agent-derived cross-task experiences to mitigate distribution shifts. This approach is theoretically grounded and tailored for **LLM agent systems**—a unique setting compared to traditional RAG or memory mechanisms.
>
> CoPS is intentionally designed to be **simple yet powerful**, enabling seamless integration into any agent framework with minimal overhead. Its ability to improve performance across tasks without the need for external databases or complex hybrid memory systems represents a **novel contribution** that bridges theoretical rigor and practical applicability.
>
> ## **Experimental Setting**
>
> Your comment about restricted experimental settings does not align with the breadth of our evaluations. CoPS was extensively tested on **diverse benchmarks** (AlfWorld, WebShop, HotPotQA), which are widely regarded as **standard and challenging** for evaluating LLM agent performance. These benchmarks span embodied reasoning, interactive environments, and multi-step QA, covering a broad range of agent capabilities.
>
> Furthermore, we performed **extensive ablation studies**, including comparisons of retrieval strategies (semantic, keyword-based, hybrid), memory constraints, and the utility of cross-task experiences. These analyses validate CoPS’ **robustness, scalability, and efficiency** in realistic scenarios. While additional benchmarks can always enhance evaluations, our experimental results already demonstrate the generalizability and effectiveness of CoPS.
>
> ## **Theoretical Analysis**
>
> CoPS is supported by a **rigorous theoretical framework** that bridges online and offline task settings. The **pessimism-based selection strategy** is grounded in principles of distributional robustness, ensuring that experience retrieval is both **utility-maximizing** and **risk-averse**.
>
> Our theoretical contributions extend beyond retrieval methods by analyzing the trade-offs between utility and safety in cross-task experience sharing. These analyses not only validate the robustness of CoPS but also highlight its theoretical novelty compared to existing approaches.
>
> ----------------
>
> We respectfully assert that CoPS makes **novel, significant contributions** by introducing a simple yet effective framework that combines theoretical rigor, broad applicability, and practical efficiency. Our evaluations are comprehensive, and our theoretical insights address critical gaps in existing paradigms. CoPS is a valuable addition to the field, advancing LLM agent systems in resource-constrained and real-world applications.
>
> We hope these clarifications address your concerns and illustrate the depth and breadth of our work. If further elaboration is required, we would be delighted to provide additional insights.
>
> Thank you again for your thoughtful review.
>
> Best regards,
> Authors of Submission #12611

---

### Author Response · Authors · 2024-11-24
**Revision Summary**

# Revision Summary

We summarize the key revisions made in our paper to address the reviewers' feedback:

---

## **Expanded Related Work**

- Added comparisons between CoPS and the O3D method, emphasizing differences in knowledge distillation and experience selection strategies, addressing **Reviewer 4n3a's Comment 1**.

---

## **Appendix Changes**

### **Appendix B**: Hyperparameter Tuning

- **Purpose**: Addressed hyperparameter impacts on performance, focusing on the effects of \(k\) (sampled experiences) and \(c\) (scaling factor).

- **Reviewer Concern**: Tackles questions from **Reviewer ct7E (Q2)** about the number of sampled experiences and their impact on performance.

---

### **Appendix D**: Quality of Demonstrations

- **Purpose**: Included ablation study comparing top-\(k\) and bottom-\(k\) trajectories to highlight the significance of high-quality demonstrations.

- **Reviewer Concern**: Added in response to **Reviewer QJKw's Comment 4**, emphasizing how demonstration quality affects performance.

---

### **Appendix E**: Experimental Robustness

- **Purpose**: Reported error bars and standard deviations for all benchmarks and models to ensure result reliability.

- **Reviewer Concern**: Responds to **Reviewer QJKw's Comment 5** about the lack of error bars and repeated experiments.

---

### **Appendix F**: Close-Sourced Models

- **Purpose**: Evaluated *CoPS* using GPT and Claude models to show its generalizability.

- **Reviewer Concern**: Directly addresses **Reviewer 4n3a's Comment 2**, demonstrating performance on state-of-the-art close-sourced models.

---

### **Appendix G**: Cross-Task Experience Impact

- **Purpose**: Compared performance with and without cross-task experiences, showing substantial improvements from sharing experiences across tasks.

- **Reviewer Concern**: Added in response to **Reviewer 4n3a's Comment 3** about the contributions of cross-task experiences.

---

### **Appendix H**: Retrieval Strategies

- **Purpose**: Performed ablation on different retrieval methods (semantic, keyword-based, hybrid) to analyze their impact on performance.

- **Reviewer Concern**: Responds to **Reviewer ct7E's Comment 4** about the lack of comparisons among retrieval strategies.

---

### **Appendix I**: Memory Constraints and Forgetting

- **Purpose**: Evaluated the performance under varying memory sizes and forgetting mechanisms to test robustness in constrained scenarios.

- **Reviewer Concern**: Tackles **Reviewer ct7E's Comment 4** about handling memory updates and forgetting mechanisms.

---

## **Figure and Presentation Improvements**

- Revised **Figure 1** with a detailed explanation of the example task to improve clarity, addressing **Reviewer ct7E's Comment 5**.

- Enhanced table formatting and captions for improved readability, including hyperparameter settings, retrieval strategies, and memory size analyses, ensuring results are easily interpretable.

---

### Meta-Review · Area_Chair_7FfG · 2024-12-18

**Metareview:**

This paper proposes a RAG-like approach, with multiple tasks. Reviewers generally commented that the work lacked significant novelty compared to the original RAG, while the experiments lacked rigor. There was a healthy discussion which was not sufficient for the reviewers to increase their scores, as concerns remained. The paper therefore fails to meet the bar for acceptance.

**Additional Comments On Reviewer Discussion:**

The main points were regarding the novelty vs. RAG and then the lack of rigor in the experiments, with simple tasks and only one seed. The authors claimed the novelty was indeed there, while saying that new benchmarks will be included in the camera ready. This seems insufficient since the paper needs to be evaluated as-is, and has not substantially changed since the initial submission.

---

### Decision · Program_Chairs · 2025-01-22

Reject